# TDFormer: A Top-Down Attention-Controlled Spiking Transformer

## Abstract

Traditional spiking neural networks (SNNs) can be viewed as a combination of multiple subnetworks with each running for one time step, where the parameters are shared, and the membrane potential serves as the only information link between them. However, the implicit nature of the membrane potential limits its ability to effectively represent temporal information. As a result, each time step cannot fully leverage information from previous time steps, seriously limiting the model's performance. Inspired by the top-down mechanism in the brain, we introduce TDFormer, a novel model with a top-down feedback structure that functions hierarchically and leverages high-order representations from earlier time steps to modulate the processing of low-order information at later stages. The feedback structure plays a role from two perspectives: 1) During forward propagation, our model increases the mutual information across time steps, indicating that richer temporal information is being transmitted and integrated in different time steps. 2) During backward propagation, we theoretically prove that the feedback structure alleviates the problem of vanishing gradients along the time dimension. We find that these mechanisms together significantly and consistently improve the model performance on multiple datasets. In particular, our model achieves state-of-the-art performance on ImageNet with an accuracy of 86.83%.

## 1 Introduction

Spiking Neural Networks (SNNs) are more energy-efficient and biologically plausible than traditional artificial neural networks (ANNs) [1]. Transformer-based SNNs combine the architectural advantages of Transformers with the energy efficiency of SNNs, resulting in a powerful and efficient models that have attracted increasing research interest in recent years [2, 3, 4, 5, 6]. However, there is still a big performance gap between existing SNNs and ANNs. This is because SNNs represent information using binary spike activations, whereas ANNs use floating-point numbers, resulting in reduced representational capacity and degraded performance. Moreover, the non-differentiability of spikes hinders effective training with gradient-based methods.

In traditional SNNs, a common approach to increase representational capacity is to expand the time step $T$. However, SNNs trained with direct coding and standard learning methods [7] lack structural mechanisms for temporal adaptation. Temporal information is solely conveyed through membrane potential dynamics, while the network architecture, parameters, and inputs remain fixed across time steps. This reliance on membrane dynamics imposes two fundamental limitations. First, temporal information can only be expressed when spikes are fired, yet firing rates are typically low across layers, restricting the bandwidth of information flow. Moreover, the cumulative nature of membrane potentials leads to loss of temporal detail, as earlier spike patterns are summed. Second, temporal gradients must propagate solely through membrane potentials, which can result in vanishing

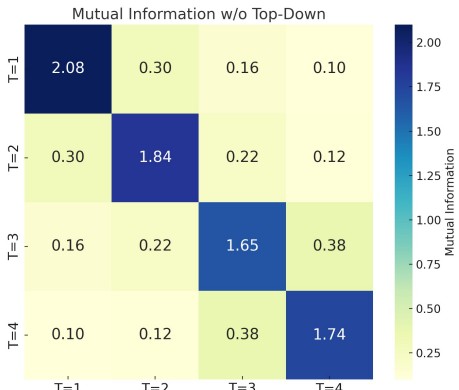
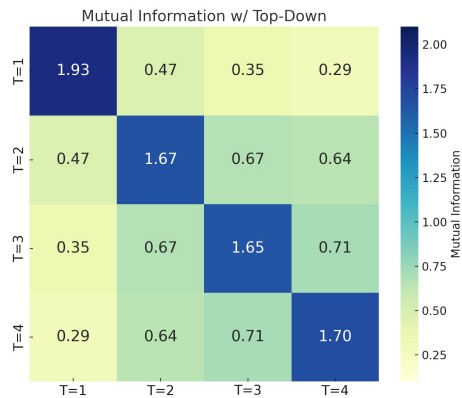

Figure 1: Visualization of mutual information matrices of features across time steps on ImageNet. The left panel shows the baseline model; the right panel shows the model incorporating feedback connections. A higher level of mutual information suggests that the model captures more consistent and temporally dependent features across time steps

gradients[8, 9]. We further confirm these limitations through temporal correlation analysis shown in Figure 1, which demonstrates the limited representational capacity of membrane potentials, and theoretical derivation in appendix B.3.

Previous work has been done to enhance the ability of SNNs to represent temporal information, e.g., by initializing the membrane potential and altering the surrogate gradients and dynamics equations [10, 11, 12]. Furthermore, some approaches have incorporated the dimension of time into attention mechanisms, resulting in time complexity that scales linearly with the number of simulation time steps [13]. However, structural mechanisms to facilitate information flow across multiple time steps remain largely unexplored. We argue that adding connections between different time steps has the following two benefits: First, in forward propagation, such connections help the model better leverage features from previous time steps. Second, in backpropagation, structural connections support gradient flow and help mitigate vanishing gradients caused by the membrane potential dynamics.

While traditional SNNs rely on bottom-up signal propagation, top-down mechanisms are prevalent in the brain, especially between the prefrontal and visual cortices [14, 15, 16, 17], as shown in Figure 2. These mechanisms are fundamental to how the brain incrementally acquires visual information over time, with higher-level cognitive processes guiding the extraction of lower-level sensory features, and prior knowledge informing the interpretation and refinement of new sensory input. Inspired by top-down mechanisms, we introduce TDFormer, a Transformer-based SNN architecture that incorporates a top-down feedback structure to improve temporal information utilization. Our main contributions can be summarized as follows:

- We identify structural limitations in traditional SNNs, showing that features across time steps exhibit weak mutual information, indicating insufficient temporal integration and utilization.

- We propose TDFormer, a Transformer-based SNN with a novel top-down feedback structure. We show that the proposed structure improves temporal information utilization, and provide theoretical analysis showing it mitigates vanishing gradients along the temporal dimension.

- We demonstrate state-of-the-art performance across multiple benchmarks with minimal energy overhead, achieving ANN-level accuracy on ImageNet while preserving the efficiency of SNNs.

## 2 Related Works

### 2.1 Transformer-based SNNs

Spikformer [2] presented the first Transformer architecture based on SNNs, laying the groundwork for spike-based self-attention mechanisms. Spike-driven TransformerV1 [5] introduced a spike-driven

mechanism to effectively process discrete-time spike signals and employed stacked transformer layers to capture complex spatiotemporal features. Built on [5], Spike-driven TransformerV2 [6] enhanced the spike-driven mechanism and added dynamic weight adjustment to improve adaptability and accuracy in processing spike data. SpikformerV2 [18] was specifically optimized for high-resolution image recognition tasks, incorporating an improved spike encoding method and a multi-layer self-attention mechanism. SpikeGPT [19] proposed an innovative combination of generative pre-trained Transformers with SNNs. SGLFormer [20] enhanced feature representations by effectively capturing both global context and local details.

## 2.2 Models with Top-Down Mechanisms

Unlike bottom-up processes that are driven by sensory stimuli, top-down attention is governed by higher cognitive processes such as goals, previous experience, or prior knowledge[21]. This mechanism progressively acquires information by guiding the focus of attention to specific regions or features of the visual scene. It can be seen as a feedback loop where higher-level areas provide signals that modulate the processing of lower-level sensory inputs, ensuring that the most relevant information is prioritized.

Many works have explored top-down attention mechanisms to improve model performance in traditional ANNs. For example, Zheng et al. [21] proposed FBTP-NN, which integrates bottom-up and top-down pathways to enhance visual object recognition, where top-down expectations modulate neuron activity in lower layers [21]. Similarly, Anderson et al. introduced a model combining bottom-up and top-down attention for image captioning and visual question answering, where top-down attention weights features based on task context [22]. Shi et al. introduced a top-down mechanism for Visual Question Answering (VQA), where high-level cognitive hypotheses influence the focus on relevant scene parts [23]. Finally, Abel and Ullman proposed a network that combines back-propagation with top-down attention to adjust gradient distribution and focus on important features [24].

## 3 Preliminaries

### 3.1 The Spiking Neuron

The fundamental distinction between SNNs and ANNs lies in their neuronal activation mechanisms. Drawing on established research [2, 4, 5, 3], we select the Leaky Integrate-and-Fire (LIF) [25] neuron model as our primary spike activation unit. LIF neuron dynamics can be formulated by:

$$V[t] = H[t](1 - S[t]) + V_{\text{reset}}S[t], \tag{1}$$

$$H[t] = V[t-1] + \frac{1}{\tau}(X[t] - (V[t-1] - V_{\text{reset}})), \tag{2}$$

$$S[t] = \Theta(H[t] - V_{\text{th}}), \tag{3}$$

where $V_{\text{reset}}$ is the reset potential. When a spike is generated, $S[t] = 1$, the membrane potential $V[t]$ is reset to $V_{\text{reset}}$; otherwise, it remains at the hidden membrane potential $H[t]$. Moreover, $\tau$ represents the membrane time constant, and the input current $X[t]$ is decay-integrated into $H[t]$.

### 3.2 Spike-Based Self-Attention Mechanisms

A critical challenge in designing spike-based self-attention is eliminating floating-point matrix multiplication in Vanilla Self-Attention (VSA) [26], which is crucial for utilizing the additive processing characteristics of SNNs.

**Spiking Self-Attention** (SSA) Zhou et al. [2] first leveraged spike dynamics to replace the softmax operation in VSA, thereby avoiding costly exponential and division calculations, and reducing energy consumption. The process of SSA is as follows:

$$I_s = \mathcal{SN}(BN(XW_I)), I \in \{Q, K, V\}, \tag{4}$$

$$\text{SSA}(Q_s, K_s, V_s) = \mathcal{SN}(Q_s K_s^\top V_s * s), \tag{5}$$

where $W \in \mathbb{R}^{T \times N \times D}$ denotes a learnable weight matrix, $I_s$ represents the spiking representations of query $Q_s$, key $K_s$, and value $V_s$. Here, $\mathcal{SN}(\cdot)$ denotes the LIF neuron, and $s$ is a scaling factor.

**Spike-Driven Self-Attention** (SDSA) Yao et al. [5, 6] improved the SSA mechanism by replacing the matrix multiplication with the Hadamard product and computing the attention via column-wise summation, effectively utilizing the additive properties of SNNs. The first version of SDSA [5] is as follows:

$$\text{SDSA}_1(Q_s, K_s, V_s) = Q_s \otimes \mathcal{SN}(\text{SUM}_c(K_s \otimes V_s)), \tag{6}$$

where $\otimes$ denotes the Hadamard product, $\text{SUM}_c(\cdot)$ represents the column-wise summation. Furthermore, the second version of SDSA [6] is described as follows:

$$\text{SDSA}_2(Q_s, K_s, V_s) = \mathcal{SN}_s((Q_s K_s^\top)V_s), \tag{7}$$

where $\mathcal{SN}_s$ denotes a spiking neuron with a threshold of $s \cdot V_{\text{th}}$. **Q-K Attention** (QKA) The work in [3] reduces the computational complexity from quadratic to linear by utilizing only the query and key. QKA can be further divided into two variants: Q-K Token Attention (QKTA) and Q-K Channel Attention (QKCA). The formulations for QKTA and QKCA are provided in Equations 8 and 9, respectively:

$$\text{QKTA}(Q_s, K_s) = \mathcal{SN}(\sum_{i=0}^{D} Q_s(i,j)) \otimes K_s, \tag{8}$$

$$\text{QKCA}(Q_s, K_s) = \mathcal{SN}(\sum_{j=0}^{N} Q_s(i,j)) \otimes K_s, \tag{9}$$

where $N$ denotes the token number, $D$ represents the channel number.

## 4 Method

In this section, we introduce TDFormer, a Transformer-based SNN model featuring a top-down feedback structure. We describe its architecture, including the division into sub-networks for feedback processing. We theoretically show that the attention module prior to the LIF neuron in the feedback pathway exhibits lower variance compared to SSA and QKTA, and we provide guidance for hyperparameter selection. Finally, we introduce the training loss and inference process. Detailed mathematical derivations are provided in appendix B.

### 4.1 TDFormer Architecture

This work is based on three backbones: SpikformerV1 [2], Spike-driven TransformerV1 [5] and QKformer [3]. These can be summarized into a unified structure, as shown in Figure 2, which consists of $L_c$ Conv-based SNN blocks, $L_t$ Transformer-based SNN blocks, and a classification head (CH). Additionally, the Transformer-based SNN blocks incorporate spike-based self-attention modules and Multi-Layer Perceptron (MLP) modules.

Apart from the backbone structure, the TDFormer architecture specifically introduces a top-down pathway called TDAC that includes two modules: the control module (CM) and the processing module (PM), as shown in Figure 2.

Viewing traditional SNNs as a sequence of $T = 1$ sub-networks with shared parameters and temporal dynamics governed by membrane potentials, we propose two approaches to introducing the top-down pathway. The first adds recurrent feedback connections between these fine-grained $T = 1$ sub-networks, enabling temporal context to propagate backward through time. The second adopts a coarser temporal resolution by dividing a sequence (e.g., $T = 4$) into fewer segments (e.g., two $T = 2$ blocks). Importantly, the additional power overhead introduced by both schemes remains minimal. Detailed analysis of power consumption is provided in appendix C.1. Both approaches can be expressed in the following unified formulation:

$$H_1 = \text{F}_{\text{tr}}\left(\text{CM}\left(S_{bu}^{(1)}, \varnothing\right)\right) \qquad H_1 \in \{0,1\}^{T \times N \times C}, S_{bu}^{(1)} \in \{0,1\}^{T \times H \times W \times C} \tag{10}$$

$$S_{td}^{(1)} = \text{PM}(H_1) \qquad S_{td}^{(1)} \in \{0,1\}^{T \times N \times C}, H_1 \in \{0,1\}^{T \times N \times C} \tag{11}$$

$$H_n = \text{F}_{\text{tr}}\left(\text{CM}\left(S_{bu}^{(n)}, S_{td}^{(n-1)}\right)\right) \qquad S_{bu}^{(n)} \in \{0,1\}^{T \times H \times W \times C}, n = 1 \ldots N \tag{12}$$

$$S_{td}^{(n)} = \text{PM}(H_n) \qquad S_{td}^{(n)} \in \{0,1\}^{T \times N \times C}, n = 1 \ldots N \tag{13}$$

$$O_n = \text{CH}(H_n) \qquad O_n \in \{0,1\}^{T \times L}, H_n \in \{0,1\}^{T \times N \times C}, n = 1 \ldots N \tag{14}$$

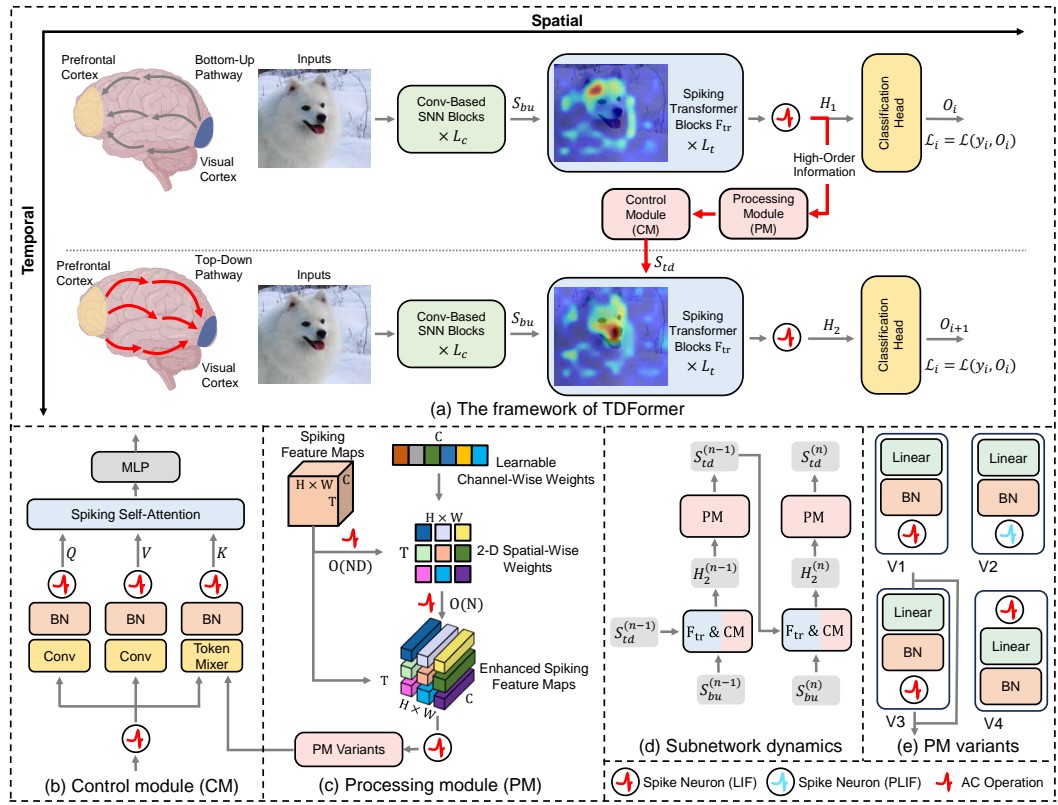

Figure 2: Overview of the TDFormer architecture. (a) Overall design inspired by top-down pathways in the brain, mimicking feedback from the prefrontal cortex to the visual cortex for temporal modulation in SNNs; (b) and (c) Detailed structures of the processing and control modules; (d) Information flow within the subnetwork, highlighting processing of feedback signals; (e) Four processing module variants, labeled v1–v4.

In the above formulation, $S_{bu}^{(n)}$ denotes the bottom-up input at time step $n$, while $S_{td}^{(n-1)}$ represents the top-down feedback from the previous step. CM is a control module that integrates bottom-up and top-down signals, and $F_{tr}$ denotes the Transformer-based processing unit. The processing module PM generates the current feedback signal $S_{td}^{(n)}$ from the high-level representation $H_n$, and CH maps $H_n$ to the final output $O_n$, where $N$ denotes the number of sub-networks. The bottom part of Figure 2 illustrates the feedback information flow between sub-networks.

**For the control module (CM),** CM derives the query $Q$, key $K$, and value $V$ vectors from the bottom-up information $S_{bu}$ and the top-down information $S_{td}$. In more detail, $S_{td}$ facilitates attention correction by controlling the attention map. The CM can be formulated as follows:

$$Q, K, V = CM(S_{bu}, S_{td}), \tag{15}$$

$$K = \mathcal{SN}(\mathrm{BN}(\mathrm{TokenMix}\left((S_{bu}, S_{td})\right))), \tag{16}$$

$$Q = \mathcal{SN}(\mathrm{BN}(\mathrm{Linear}(S_{bu}))), V = \mathcal{SN}(\mathrm{BN}(\mathrm{Linear}(S_{bu}))). \tag{17}$$

We choose concatenation along the channel dimension as the default token mixer, which allows us to combine the features of the current time step with those from previous time steps, and use the fused information to dynamically adjust the self-attention map. After passing through the CM, the query $Q$, key $K$ and value $V$ vectors are fed into the self-attention module to obtain the top-down attention map. To prevent the fusion of top-down information from altering the distribution of $K$ in the self-attention computation, we first normalize the combined features, and then apply spike discretization before computing self-attention. Ablation studies on different CM variants are provided in the appendix C.2.

**The processing module (PM)** PM includes both channel-wise token mixer and spatial-wise token mixer [27]. The feature enhancement component enhances the original spiking feature maps $\mathbf{X}$ by learning channel-wise $\mathbf{W}_c$ and computing spatial-wise attention maps $\mathbf{M}_{\text{spatial}}$. This attention mechanism requires very few parameters and has a time complexity of $O(ND)$. This operation is represented as:

$$\mathbf{M}_{\text{spatial}}(t, n) = \sum_{c=1}^{C} \mathbf{W}_c \cdot \mathbf{X}_{t,n,c}, \tag{18}$$

$$\mathbf{M}_{\text{spatial}} = \text{clamp}\left(\mathbf{M}_{\text{spatial}}, b, a\right). \tag{19}$$

where $\mathbf{X}_{t,n,c}$ represents the spiking activation at time $t$, spatial position $n$ (corresponding to the 2D coordinate $(h, w)$ in the feature map), and channel $c$. Here, $a$ and $b$ are hyperparameters. We theoretically derive their effects on the PM output, and the details are given in appendix B.2. The spatial attention map $\mathbf{M}_{\text{spatial}}$ weights the spiking feature map $\mathbf{X}$ via element-wise multiplication, with broadcasting over the channel dimension:

$$\mathbf{O} = \mathcal{SN}(\mathbf{X} \odot \mathbf{M}_{\text{spatial}}). \tag{20}$$

The attention embedding spaces are different across layers, and we aim to use a PM variants to align the top-down information with the embedding spaces of different layers. We explored four PM variants that serve as the channel-wise token mixer, which are illustrated in Figure 2.

We introduce a clamp operation in the attention module to enforce a strict upper bound on the variance of the attention map which is formally stated in Proposition 4.1. Excessive variance can lead to gradient vanishing, as gradients in spiking neurons are only generated near the firing threshold of the membrane potential. Outside this narrow region, the gradient tends to vanish. Furthermore, high variance may introduce outliers, resulting in significant quantization errors during spike generation. The effect of the clamp operation on the gradient is shown in the Figure appendix C.2.

**Proposition 4.1.** *The upper bound* $\overline{Var}(Y_{tnc})$ *for the* $\mathbf{X} \odot \mathbf{M}_{spatial}$ *is given as follows:*

$$\overline{Var}(Y_{tnc}) = \begin{cases} a^2(f^2 - f + \frac{1}{2}) + ab(1 - 2f) + \frac{b^2}{2}, & \text{if } 0 \leq f \leq \frac{a+b}{2a}, \\ \frac{a^2 + 2ab + b^2 - 4fab}{4}, & \text{if } \frac{a+b}{2a} \leq f \leq 1, \end{cases} \tag{21}$$

*where we assume each* $\mathbf{X}_{t,n,c}$ *is independent random variable* $X_{tnc} \sim Bernoulli(f)$, *with* $f$ *as the firing rate.*

Additionally, the clamp operation eliminates the need for scaling operations in attention mechanisms (e.g., QK product scaling), simplifying computations, reducing complexity, and improving energy efficiency in hardware implementations. The detailed proofs of this proposition are provided in appendix B.1.

## 4.2 Loss Function

The loss of the TDFormer can be formulated as follows:

$$\mathcal{L}_{\text{TDFormer}} = \sum_{n=1}^{N} \alpha_n \mathcal{L}(y, O_n), \quad \sum_{n=1}^{N} \alpha_n = 1, \quad 0 \leq \alpha_n \leq 1. \tag{22}$$

Here, $\alpha_n$ are hyperparameters. To maintain the overall loss scale, we apply a weighted average over the losses from all $N$ stages, assigning a larger weight to the final output loss. This is because we believe that the receptive field in the temporal dimension increases as time progresses. Since the earlier stages lack feedback from future steps, their outputs are less accurate and thus subject to weaker supervision. By contrast, the final stage benefits from a larger temporal receptive field due to feedback, making its output more reliable. Therefore, during testing, only the output from the last sub-network is used for evaluation.

## 4.3 Top-down feedback enhances temporal dependency

Top-down feedback enhances temporal dependency from two perspectives. First, from the forward propagation perspective, we compute the mutual information matrix between features at different time

Table 1: Comparison with the baseline and previous work on ImageNet. The result in bold indicates superior performance compared to the baseline. SOTA is marked with *, previous SOTA with #. The default PM variant is v1.

| Methods | Spike | Architecture | ImageNet | | | |
| --- | --- | --- | --- | --- | --- | --- |
| | | | Time Step | Power (mJ) | Param (M) | Acc (%) |
| ViT [28] | ✗ | ViT-B/16($384^2$) | 1 | 254.84 | 86.59 | 77.90 |
| DeiT [29] | ✗ | DeiT-B($384^2$) | 1 | 254.84 | 86.59 | 83.10 |
| Swin [30] | ✗ | Swin Transformer-B($384^2$) | 1 | 216.20 | 87.77 | 84.50 |
| Spikingformer [4] | ✓ | Spikingformer-8-768 | 4 | 13.68 | 66.34 | 75.85 |
| SpikformerV1 [2] | ✓ | Spikformer-8-512 | 4 | 11.58 | 29.68 | 73.38 |
| | ✓ | Spikformer-8-768 | 4 | 21.48 | 66.34 | 74.81 |
| SDTV2 [6] | ✓ | Meta-SpikeFormer-8-384 | 4 | 32.80 | 31.30 | 77.20 |
| | ✓ | Meta-SpikeFormer-8-512 | 4 | 52.40 | 55.40 | 80.00 |
| E-Spikeformer [31] | ✓ | E-Spikeformer | 8 | 30.90 | 83.00 | 84.00 |
| | ✓ | E-Spikeformer | 8 | 54.70 | 173.00 | 85.10 |
| | ✓ | E-Spikeformer | 8 | - | 173.00 | 86.20 # |
| QKFormer [3] | ✓ | HST-10-768 ($224^2$) | 4 | 38.91 | 64.96 | 84.22 |
| | ✓ | HST-10-768 ($288^2$) | 4 | 64.27 | 64.96 | 85.20 |
| | ✓ | HST-10-768 ($384^2$) | 4 | 113.64 | 64.96 | 85.65 |
| TDFormer | ✓ | HST-10-768 ($224^2$) | 4 | 38.93 | 65.55 | **85.37(+1.15)** |
| | ✓ | HST-10-768 ($288^2$) | 4 | 64.39 | 65.55 | **86.29(+1.09)** |
| | ✓ | HST-10-768 ($224^2$) | 4 | 39.10 | 69.09 | **85.57(+1.35)** |
| | ✓ | HST-10-768 ($288^2$) | 4 | 64.45 | 69.09 | **86.43 (+1.23)** |
| | ✓ | HST-10-768 ($384^2$) | 4 | 113.79 | 69.09 | **86.83 (+1.18)*** |

steps, as shown in Figure 1. Second, from the backward propagation perspective, we demonstrate that introducing top-down feedback helps alleviate the problem of vanishing gradients along the temporal dimension. We present the following theorem:

**Definition 4.2.** $\epsilon^l(t)$ is defined as the sensitivity of the membrane potential $\mathbf{H}^l(t+1)$ to its previous state $\mathbf{H}^l(t)$, and is computed as:

$$\epsilon^l(t) \equiv \frac{\partial \mathbf{H}^l(t+1)}{\partial \mathbf{H}^l(t)} + \frac{\partial \mathbf{H}^l(t+1)}{\partial \mathbf{S}^l(t)} \frac{\partial \mathbf{S}^l(t)}{\partial \mathbf{H}^l(t)}, \tag{23}$$

where $l$ indexes the layer.

**Theorem 4.3.** *We adopt the rectangular function as the surrogate gradient, following the setting used in previous studies[8, 9, 12]. For a conventional SNN, the sensitivity of the membrane potential is expressed as follows:*

$$\epsilon^l(t)_{jj} = \begin{cases} 0, & \frac{1}{2}\vartheta < H_j^l(t) < \frac{3}{2}\vartheta, \\ 1 - \frac{1}{\tau}, & otherwise. \end{cases} \tag{24}$$

*For SNN with top-down feedback structure, the sensitivity of the membrane potential can be expressed as:*

$$\epsilon^l(t)_{jj} = \begin{cases} \frac{\partial \varphi_\theta(\mathbf{S}^l(t))}{\partial \mathbf{S}^l(t)}, & \frac{1}{2}\vartheta < H_j^l(t) < \frac{3}{2}\vartheta, \\ 1 - \frac{1}{\tau}, & otherwise. \end{cases} \tag{25}$$

*where $\vartheta$ is the spike threshold, $\tau$ is a time constant and $\varphi_\theta$ is a differentiable feedback function parameterized by $\theta$.*

According to Equation 24, $\epsilon^l(t)$ becomes zero within an easily-reached interval, and outside that interval, it is upper-bounded by a small value $1 - \frac{1}{\tau}$, since $\tau$ is typically close to 1 in practice[32, 33, 34, 9]. In contrast, our method allows non-zero gradients within this interval, and the $\frac{\partial \varphi_\theta(\mathbf{S}^l(t))}{\partial \mathbf{S}^l(t)}$ can

Table 2: Comparison with the baselines and previous work on static datasets: CIFAR-10 and CIFAR-100. Conventions align with those in Table 1. The default PM variant is v1.

| Methods [Architecture] | Time Step | CIFAR-10 Acc (%) | CIFAR-100 Acc (%) |
|---|---|---|---|
| STBP-tdBN [33] [ResNet-19] | 4 | 92.92 | 70.86 |
| TET [32] [ResNet-19] | 4 | 94.44 | 74.47 |
| SDTV1[5][SDT-2-512] | 4 | 95.60 | 78.40 |
| QKformer [3] [HST-4-384] | 4 | 96.18 # | 81.15 # |
| SpikformerV1 [2] [Spikformer-4-384] | 2 | 93.59 | 76.28 |
| | 4 | 95.19 | 77.86 |
| SpikformerV1(ours)[Spikformer-4-384] | 2 | 93.65 | 75.29 |
| | 4 | 94.73 | 77.88 |
| TDFormer[Spikformer-4-384] | 2 | **94.17 (+0.52)** | **75.79 (+0.50)** |
| | 4 | **95.11 (+0.38)** | **77.99 (+0.11)** |
| SDTV1(ours)[SDT-2-256] | 4 | 94.47 | 76.05 |
| SDTV1(ours)[SDT-2-512] | 4 | 95.78 | 79.15 |
| TDFormer[SDT-2-256] | 4 | **94.61 (+0.14)** | **76.23 (+0.18)** |
| TDFormer[SDT-2-512] | 4 | **96.07 (+0.29)** | **79.67 (+0.52)** |
| TDFormer [HST-4-384] | 4 | **96.51 (+0.33)\*** | **81.45 (+0.30)\*** |

exceed $1 - \frac{1}{\tau}$. This property helps to alleviate the vanishing gradient problem along the temporal dimension. The detailed proof is provided in the appendix B.3.

## 5 Experiments

We evaluate our models on several datasets: CIFAR-10 [35], CIFAR-100 [35], CIFAR10-DVS [36], DVS128 Gesture [37], ImageNet [38], CIFAR-10C [39] and ImageNet-C [39]. For the smaller datasets, we employ the feedback pathway on SpikformerV1 [2] , Spike-driven TransformerV1 [5] and QKformer[3], experimenting with different configurations tailored to each dataset. For the large-scale datasets, we utilize QKformer[3] as baselines. Specific implementation details are provided in appendix A.

### 5.1 Experiments on ImageNet

Table 1 presents the results for the large-scale dataset ImageNet. The incorporation of top-down feedback structure has demonstrated significant improvements on E-spikformer, which is the previous SOTA model of SNNs. Notably, compared to QKFormer, increasing the model size by merely 0.02 million parameters and 0.59 millijoules of power consumption leads to a significant gain of 1.15% in top-1 accuracy on the ImageNet dataset. Our model sets a new SOTA performance in the SNN field. This milestone lays a solid foundation for advancing SNNs toward large-scale networks, further bridging the gap between SNNs and traditional deep learning models. Furthermore, we calculate the power of TDFormer following the method in [3], as detailed in Table 1. TDFormer results in a slight increase in energy consumption due to the feedback structure, but it achieves superior performance with minimal additional power usage. The detailed calculation of power consumption is provided in the appendix C.1.

### 5.2 Experiments on Neuromorphic and CIFAR Datasets

Table 3 presents the results for the neuromorphic datasets CIFAR10-DVS and DVS128 Gesture. Our proposed TDFormer consistently outperforms the baselines across all experiments, except for the Spiking Transformer-2-256 at a time step of 10. Furthermore, we achieve SOTA results, with an accuracy of 85.83% on CIFAR10-DVS using the HST-2-256 (V1), marking a notable improvement

Table 3: Comparison with the baselines and previous work on the Neuromorphic Dataset. Conventions align with those in Table 1. The default PM variant is v1.

| Methods [Architecture] | CIFAR10-DVS | | DVS128 Gesture | |
|---|---|---|---|---|
| | Time Step | Acc (%) | Time Step | Acc (%) |
| STBP-tdBN [33] [ResNet-19] | 10 | 67.80 | 40 | 96.90 |
| DSR [40] [VGG-11] | 10 | 77.30 | - | - |
| SDTV1 [5][SDT-2-256] | 16 | 80.00 | 16 | 99.30 # |
| SpikformerV1 [2] [Spikformer-2-256] | 10 | 78.90 | 10 | 96.90 |
| | 16 | 80.90 | 16 | 98.30 |
| Spikingformer [4] [Spikingformer-2-256] | 10 | 79.90 | 10 | 96.20 |
| | 16 | 81.30 | 16 | 98.30 |
| Qkformer [3] [HST-2-256] | 16 | 84.00 # | 16 | 98.60 |
| SpikformerV1(ours) [Spikformer-2-256] | 10 | 78.08 | - | - |
| | 16 | 79.40 | - | - |
| TDFormer [Spikformer-2-256] | 10 | **78.90 (+0.82)** | - | - |
| | 16 | **81.70 (+2.30)** | - | - |
| SDTV1(ours) [SDT-2-256] | 10 | 75.22 | 10 | 96.79 |
| | 16 | 77.07 | 16 | 97.98 |
| TDFormer[SDT-2-256] | 10 | 75.05 (-0.17) | 10 | **96.92 (+0.13)** |
| | 16 | **77.45 (+0.38)** | 16 | **99.65 (+1.67)*** |
| TDFormer[HST-2-256] | 16 | **85.83 (+1.83)*** | 16 | **98.96 (+0.36)** |

of 1.83% compared to the previous SOTA model, QKformer. We also achieve 99.65% accuracy on DVS128 Gesture using the Spiking Transformer-2-256 (V1) at 16 time steps.

In addition, the results for the static datasets CIFAR-10 and CIFAR-100 are summarized in Table 2. Compared to the baselines, the proposed TDFormer consistently demonstrates significant performance improvements across all experiments, with the exception of Spikformer-4-384 (V1) at time step 6. Furthermore, we achieve the SOTA performance, attaining 96.51% accuracy on CIFAR-10 and 81.45% on CIFAR-100 using the HST-2-256 (V1) at a time step of 4.

### 5.3 Model Generalization Analysis

As reported in Table 5, we report results averaged over five random seeds for reliability. Our model consistently improves performance across time steps and depths. To assess robustness, we evaluate on the CIFAR-10C dataset with 15 corruption types. As shown in Table 7, the model equipped with the TDAC module consistently achieves higher accuracy under various distortion settings.

Moreover, we provide a visualization analysis of the TDFormer attention modules on CIFAR-10C and ImageNet-C. The specific results can be seen in Figure 4 and Figure 5 of the appendix C. We find that after adding the TDAC module, the model focuses more on the targets and their surrounding areas. This indicates that TDAC can filter noise and irrelevant information, allowing the model to focus more on task-related information.

## 6 Conclusion

In this study, we propose TDFormer, which integrates an adaptive top-down feedback structure into Transformer-based SNNs, addressing a key limitation of temporal information utilization in existing models by incorporating biological top-down mechanisms. The TDFormer model outperforms traditional Transformer-based SNNs, achieving SOTA performance across all evaluated datasets. Our work suggests that the top-down feedback structure could be a valuable component for Transformer-based SNNs and offers insights for future research into more advanced, biologically inspired neural architectures that better mimic human cognition.

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

# A  Implementation Details

## A.1  Training Protocols

We adopted the following training protocols:

- **Spike Generation**: We used a rate-based method for spike generation [2].

- **Data Augmentation and Training Duration**: SpikformerV1 experiments followed [2], while Spike-driven TransformerV1 experiments followed [5], furthermore QKformer experiments followed the experimental setting in and [3].

- **Optimization**: We employed AdamW [41] as the optimizer for our experiments. The learning rate was set to $3 \times 10^{-4}$ for the Spike-driven TransformerV1. For SpikformerV1, we used a learning rate of $5 \times 10^{-4}$ on static datasets and $1 \times 10^{-3}$ on neuromorphic datasets. Additionally, we utilized a cosine learning rate scheduler to adjust the learning rate dynamically during training. Specifically, for QKformer, we fine-tuned the pretrained network with a base learning rate of $2 \times 10^{-5}$ for 15 epochs, due to the high cost of direct training on ImageNet using 4 time steps.

- **Batch Size**: The batch sizes for different datasets and models are specified in Table 4.

Table 4: Batch sizes for different datasets and models.

| Dataset | Model | Batch Size |
|---|---|---|
| CIFAR-10 and CIFAR-100 | SpikeformerV1 Spike-driven TransformerV1 | 128 64 |
| CIFAR10-DVS and DVS128 Gesture | SpikeformerV1 Spike-driven TransformerV1 | 16 16 |
| ImageNet | QKformer | 57 |

## A.2  Datasets

Our experiments evaluated the performance and robustness of the TDFormer model using the following datasets:

- **CIFAR-10:** This dataset contains 60,000 $32 \times 32$ color images divided into 10 classes [35].

- **CIFAR-100:** This dataset is similar to CIFAR-10 but includes 100 classes, providing a more challenging classification task [35].

- **CIFAR10-DVS:** This is an event-based version of the CIFAR-10 dataset [36].

- **DVS128 Gesture:** This is an event-based dataset for gesture recognition with 11 classes [37].

- **ImageNet:** This large-scale dataset contains over 1.2 million images divided into 1,000 classes [38].

- **CIFAR-10C:** This is a corrupted version of CIFAR-10 with 19 common distortion types, used to assess robustness [39].

- **ImageNet-C:** This dataset is a corrupted version of ImageNet, designed similarly to CIFAR-10C [39].

## A.3  Computational Environment

### A.3.1  Software Setup

We utilized PyTorch version 2.0.1 with CUDA 11.8 support and SpikingJelly version 0.0.0.0.12 as the primary software tools.

 **A.3.2 Hardware Setup.**

430 For the smaller dataset experiments, we utilized the following configuration:

431 • **Hardware Used:** NVIDIA L40S and L40 GPUs.

432 • **Configuration:** Single-GPU for each experiment.

433 • **Memory Capacity:** Each GPU is equipped with 42 GB of memory.

434 For the large-scale dataset (ImageNet) experiments, we employed the following setup:

435 • **Hardware Used:** NVIDIA H20 GPUs.

436 • **Configuration:** Eight-GPU for each experiment.

437 • **Memory Capacity:** Each GPU provides 96 GB of memory.

438 **A.4 Random Seed**

439 To ensure the comparability of the results, we selected the same random seeds as those in the baseline
440 paper. To ensure robustness, we also conducted experiments with random seeds 0, 42, 2024, 3407
441 and 114514, averaging the results. Detailed results are presented in Table 5.

442 # B Mathematical Derivations

443 **B.1 Detailed proofs of the upper bound on PM output variance**

444 *Proof.* We assume that each $\mathbf{M}_{\text{spatial}}(t, n)$ is an independent random variable $M_{tn}$. Given that
445 $b \leq M_{tn} \leq a$, it follows that $b \leq \mathbb{E}[M_{tn}] \leq a$. Furthermore, when $X_{tnc} \neq 0$, we have:

$$(X_{tnc}M_{tn} - b)(a - X_{tnc}M_{tn}) \geq 0, \tag{26}$$

446 which expands to:

$$-(X_{tnc}M_{tn})^2 + (a + b)(X_{tnc}M_{tn}) - ab \geq 0. \tag{27}$$

447 Taking the expectation on both sides yields:

$$\mathbb{E}\left[(X_{tnc}M_{tn})^2\right] \leq (a + b)\mathbb{E}\left[X_{tnc}M_{tn}\right] - ab. \tag{28}$$

448 Using the Law of Total Variance, we can decompose the variance of $Y_{tnc}$ as:

$$\text{Var}(Y_{tnc}) = \mathbb{E}[\text{Var}(Y_{tnc}|X_{tnc})] + \text{Var}(\mathbb{E}[Y_{tnc}|X_{tnc}]). \tag{29}$$

449 For the first term, the expectation of the conditional variance can be expressed as:

$$\mathbb{E}[\text{Var}(Y_{tnc}|X_{tnc})] = f \cdot \text{Var}(Y_{tnc}|X_{tnc} = 1) + (1 - f) \cdot \text{Var}(Y_{tnc}|X_{tnc} = 0). \tag{30}$$

450 For the second term, the variance of the conditional expectation can be expanded as:

$$\text{Var}(\mathbb{E}[Y_{tnc}|X_{tnc}]) = \mathbb{E}[\mathbb{E}[Y_{tnc}|X_{tnc}]^2] - \mathbb{E}[\mathbb{E}[Y_{tnc}|X_{tnc}]]^2. \tag{31}$$

451 By substituting the conditional probabilities, we have:

$$\text{Var}(\mathbb{E}[Y_{tnc}|X_{tnc}]) = f \cdot \mathbb{E}[Y_{tnc}|X_{tnc} = 1]^2 - f^2 \cdot \mathbb{E}[Y_{tnc}|X_{tnc} = 1]^2. \tag{32}$$

452 Combining the two terms, the total variance becomes:

$$\text{Var}(Y_{tnc}) = f \cdot \text{Var}(Y_{tnc}|X_{tnc} = 1) + (f - f^2) \cdot \mathbb{E}[Y_{tnc}|X_{tnc} = 1]^2. \tag{33}$$

453 From Equation 32, we define $\mathbb{E}[Y_{tnc}|X_{tnc} = 1] = \mu$. Substituting this definition, the variance can be
454 rewritten as:

$$\text{Var}(Y_{tnc}) = f \cdot (\mathbb{E}[Y_{tnc}^2|X_{tnc} = 1] - \mu^2) + (f - f^2) \cdot \mu^2. \tag{34}$$

455 Using the constraints $b \leq M_{tn} \leq a$, we have the following bound for $\text{Var}(Y_{tnc}|X_{tnc} = 1)$:

$$\text{Var}(Y_{tnc}|X_{tnc} = 1) \leq (a + b)\mu - ab - \mu^2. \tag{35}$$

By substituting this into the total variance expression, the upper bound of $\mathrm{Var}(Y_{tnc})$ becomes:

$$\mathrm{Var}(Y_{tnc}) \leq f \cdot ((a+b)\mu - ab - \mu^2) + (f - f^2) \cdot \mu^2$$
$$\leq -f^2 \cdot \left(\mu - \frac{a+b}{2f}\right)^2 + \frac{a^2 + 2ab + b^2 - 4fab}{4}. \tag{36}$$

Next, we will prove that this upper bound can be achieved with equality under specific conditions.

**Case 1:** When $\frac{a+b}{2a} \leq f \leq 1$, we assume that:

$$\mathbb{E}[Y_{tnc}|X_{tnc} = 1] = \frac{a+b}{2f}, \quad M_{tn} = a \text{ or } b. \tag{37}$$

Here, $M_{tn}$ is a binary random variable, taking the value $a$ with probability $p$ and the value $b$ with probability $1-p$. Using this assumption, we can express the conditional expectation $\mathbb{E}[Y_{tnc}|X_{tnc} = 1]$ as:

$$\mathbb{E}[Y_{tnc}|X_{tnc} = 1] = pa + (1 - p)b. \tag{38}$$

Substituting $\mathbb{E}[Y_{tnc}|X_{tnc} = 1] = \frac{a+b}{2f}$ into the above equation, we solve for $p$:

$$pa + (1 - p)b = \frac{a+b}{2f} \Rightarrow p = \frac{a + b - 2bf}{2f(a - b)}. \tag{39}$$

The variance of $Y_{tnc}$ under this distribution is maximized when $M_{tn}$ follows this binary distribution. Substituting $p$ into the variance formula, the maximum variance is given by:

$$\max(\mathrm{Var}(Y_{tnc})) = \frac{a^2 + 2ab + b^2 - 4fab}{4}. \tag{40}$$

**Case 2:** When $0 \leq f \leq \frac{a+b}{2a}$, the upper bound is achieved when $M_{tn} = a$. In this scenario, $M_{tn}$ is deterministic, and therefore:

$$Y_{tnc} = X_{tnc}M_{tn} = X_{tnc}a, \quad \mathbb{E}[Y_{tnc}|X_{tnc} = 1] = a. \tag{41}$$

Substituting this into the variance formula, the maximum variance simplifies to:

$$\max(\mathrm{Var}(Y_{tnc})) = a^2(f^2 - f + 1/2) + ab(1 - 2f) + b^2/2. \tag{42}$$

The proof is now complete. $\qquad\square$

We observe that both SSA and QKTA exhibit significantly larger variance compared to our proposed attention mechanism. Their variances are expressed as follows:

**Variance of QKTA:**

$$\mathrm{Var}(\mathrm{QKTA}) = df_Q(1 - f_Q), \tag{43}$$

where $d$ is the feature dimension, and $f_Q$ represents the firing rate of the query.

**Variance of SSA:**

$$\mathrm{Var}(\mathrm{SSA}) = Nd\Big(f_Q f_K f_V(1 - f_Q)(1 - f_K)(1 - f_V)$$
$$+ f_Q f_K f_V^2(1 - f_Q)(1 - f_K)$$
$$+ f_Q f_K^2 f_V(1 - f_Q)(1 - f_V)$$
$$+ f_Q^2 f_K f_V(1 - f_K)(1 - f_V)$$
$$+ f_Q f_K^2 f_V^2(1 - f_Q)$$
$$+ f_Q^2 f_K f_V^2(1 - f_K)$$
$$+ f_Q^2 f_K^2 f_V(1 - f_V)\Big), \tag{44}$$

where $N$ is the number of spatial locations, $d$ is the feature dimension, and $f_Q, f_K, f_V$ are the firing rates of the query, key, and value.

**Comparison with Our Attention Mechanism:** The variance of QKTA scales linearly with $d$. By contrast, the variance of SSA grows with both $N$ and $d$, resulting in significantly larger values compared to QKTA. Our proposed attention mechanism is particularly effective in scenarios with large spatial ($N$) and feature ($d$) dimensions. The strict upper bound on output variance ensures numerical stability, preventing vanishing during training. Additionally, this upper bound eliminates the need for traditional scaling operations (e.g., scaling factors in QK products), simplifying computations, reducing complexity, and enhancing energy efficiency.

## B.2 The mathematical properties of hyperparameters

Next, we will analyze the expectation and variance of the PM and propose an appropriate selection of hyperparameters to ensure output stability.

**Lemma B.1.** *if the set* $\{c \in \mathbb{N} : w_c = 0\}$ *is finite and* $\exists\, m, M > 0$, $\forall\, c \in \mathbb{N}$, $m \le |w_c| \le M$, *then:*

$$w_c' = \lim_{C \to \infty} \frac{w_c}{\sqrt{\sum_{c=1}^{C} w_c^2}} = 0 \tag{45}$$

*Proof.* We begin by defining the normalized weight:

$$w_c' = \frac{w_c}{\sqrt{\sum_{c=1}^{C} w_c^2}}. \tag{46}$$

By assumption, there are $k$ terms where $w_c = 0$, and for the remaining $C - k$ terms, the weights satisfy:

$$m^2 \le w_c^2 \le M^2 \quad \text{for all } c. \tag{47}$$

Thus, the sum of squares of the weights is bounded as follows:

$$(C - k)m^2 \le \sum_{c=1}^{C} w_c^2 \le (C - k)M^2. \tag{48}$$

Taking the square root, we find that the denominator grows as:

$$\sqrt{\sum_{c=1}^{C} w_c^2} \ge \sqrt{(C - k)m^2} \sim O(\sqrt{C}). \tag{49}$$

Using the bound $|w_c| \le M$, the normalized weight $w_c'$ satisfies:

$$|w_c'| = \frac{|w_c|}{\sqrt{\sum_{c=1}^{C} w_c^2}} \le \frac{M}{\sqrt{\sum_{c=1}^{C} w_c^2}} \le \frac{M}{\sqrt{(C - k)m^2}}. \tag{50}$$

To ensure $|w_c'| < \epsilon$ for a given $\epsilon > 0$, it suffices to require:

$$\frac{M}{\sqrt{(C - k)m^2}} < \epsilon. \tag{51}$$

Rearranging, this condition can be rewritten as:

$$C \ge \frac{M^2}{m^2 \epsilon^2} + k. \tag{52}$$

As $C \to \infty$, the condition $C \ge \frac{M^2}{m^2 \epsilon^2} + k$ is always satisfied. Thus, for any $\epsilon > 0$, we have $|w_c'| < \epsilon$, which implies:

$$\lim_{C \to \infty} w_c' = 0. \tag{53}$$

The proof is complete. $\qquad\square$

**Lemma B.2.** *We assume that the features across different channels are independent and identically distributed (i.i.d.). When the number of channels $C$ is large, we have:*

$$M_{tn} \sim \mathcal{N}\left(\sum_{c=1}^{C} w'_c f_r, \sum_{c=1}^{C} w'^2_c f_r(1 - f_r)\right), \quad C \to \infty, \tag{54}$$

$$M_{tn} = \sum_{c=1}^{C} x_{tnc} w'_c. \tag{55}$$

*where $x \in X$, $x \sim Bernoulli(f_r)$, $f_r$ represents the firing rate (the probability of $x_{tnc} = 1$).*

*Proof.* To prove this lemma, we use the characteristic function method. The characteristic function of a Bernoulli random variable $x_{tnc}$ is given by:

$$\Phi_{x_{tnc}}(t) = \mathbb{E}\left[e^{itx_{tnc}}\right] = f_r e^{it} + (1 - f_r). \tag{56}$$

For the weighted variable $w'_c x_{tnc}$, its characteristic function is:

$$\Phi_{w'_c x_{tnc}}(t) = \mathbb{E}\left[e^{itw'_c x_{tnc}}\right] = f_r e^{itw'_c} + (1 - f_r). \tag{57}$$

Since the features across channels are independent, the characteristic function of $M_{tn}$ is:

$$\Phi_{M_{tn}}(t) = \prod_{c=1}^{C} \Phi_{w'_c x_{tnc}}(t). \tag{58}$$

Substituting the expression for $\Phi_{w'_c x_{tnc}}(t)$:

$$\Phi_{M_{tn}}(t) = \prod_{c=1}^{C} \left(f_r e^{itw'_c} + (1 - f_r)\right). \tag{59}$$

$$f_r e^{itw'_c} + (1 - f_r) = f_r\left(1 + itw'_c - \frac{1}{2}t^2 w'^2_c + o(w'^2_c)\right) + (1 - f_r)$$

$$\approx 1 + f_r(itw'_c - \frac{1}{2}t^2 w'^2_c). \tag{60}$$

Thus, the characteristic function becomes:

$$\Phi_{M_{tn}}(t) \approx \prod_{c=1}^{C} \left(1 + f_r(itw'_c - \frac{1}{2}t^2 w'^2_c)\right). \tag{61}$$

Taking the logarithm to simplify the product into a sum:

$$\ln \Phi_{M_{tn}}(t) = \sum_{c=1}^{C} \ln\left(1 + f_r(itw'_c - \frac{1}{2}t^2 w'^2_c)\right)$$

$$= \sum_{c=1}^{C} f_r itw'_c - \frac{1}{2}t^2 w'^2_c f_r + \frac{1}{2}t^2 w'^2_c f_r^2 + O(w'^2_c), \tag{62}$$

where we used $\ln(1 + x) = x - \frac{1}{2}x^2 + O(x^2)$ for small $x$.

Separating terms, we get:

$$\ln \Phi_{M_{tn}}(t) \approx it \sum_{c=1}^{C} w'_c f_r - \frac{1}{2}t^2 \sum_{c=1}^{C} w'^2_c f_r(1 - f_r). \tag{63}$$

Exponentiating the logarithm gives:

$$\Phi_{M_{tn}}(t) = \exp\left( it \sum_{c=1}^{C} w_c' f_r - \frac{1}{2} t^2 \sum_{c=1}^{C} w_c'^2 f_r (1 - f_r) \right).$$ (64)

This is the characteristic function of a normal distribution with:

$$\text{Mean:} \quad \mu = \sum_{c=1}^{C} w_c' f_r,$$ (65)

$$\text{Variance:} \quad \sigma^2 = \sum_{c=1}^{C} w_c'^2 f_r (1 - f_r).$$ (66)

Since the characteristic function corresponds to a normal distribution, we conclude:

$$M_{tn} \sim \mathcal{N}\left( \sum_{c=1}^{C} w_c' f_r, \sum_{c=1}^{C} w_c'^2 f_r (1 - f_r) \right).$$ (67)

The proof is complete. $\square$

**Lemma B.3.** *The distributions of $X_{tnc}$ and $M_{tn}$ can be considered independent when the number of channels $C$ is large. Specifically, for all $t_1, t_2 \in \mathbb{R}$, we have:*

$$\phi_{M_{tn}, X_{tnc}}(t_1, t_2) = \phi_{M_{tn}}(t_1) \cdot \phi_{X_{tnc}}(t_2), \quad C \to \infty,$$ (68)

*where $\phi_X(t)$ represents the characteristic function of $X$.*

*Proof.* The joint characteristic function of $M_{tn}$ and $X_{tnc}$ is given by:

$$\phi_{M_{tn}, X_{tnc}}(t_1, t_2) = \mathbb{E}\left[ e^{(it_1 M_{tn} + it_2 X_{tnc})} \right]$$
$$= \mathbb{E}\left[ e^{\left( it_1 \sum_c w_c' X_{tnc} + it_2 X_{tnc} \right)} \right].$$ (69)

Separating $X_{tnc}$ and the sum $\sum_{i \neq c} w_i' X_{tni}$, we rewrite:

$$\phi_{M_{tn}, X_{tnc}}(t_1, t_2) = \mathbb{E}\left[ e^{\left( it_1 \sum_{i \neq c} w_i' X_{tni} + i X_{tnc}(t_2 + t_1 w_c') \right)} \right]$$
$$= \mathbb{E}\left[ e^{\left( it_1 \sum_{i \neq c} w_i' X_{tni} \right)} \right] \cdot \mathbb{E}\left[ e^{\left( i X_{tnc}(t_2 + t_1 w_c') \right)} \right].$$ (70)

Using the independence of $X_{tni}$ across channels:

$$\phi_{M_{tn}, X_{tnc}}(t_1, t_2) = \prod_{i \neq c} \mathbb{E}\left[ e^{\left( it_1 w_i' X_{tni} \right)} \right] \cdot \mathbb{E}\left[ e^{\left( i X_{tnc}(t_2 + t_1 w_c') \right)} \right].$$ (71)

Substituting the characteristic function of Bernoulli random variables $X_{tnc} \sim \text{Bernoulli}(f)$:

$$\mathbb{E}\left[ e^{it X_{tnc}} \right)] = (1 - f) + f e^{it}.$$ (72)

Thus:

$$\phi_{M_{tn}, X_{tnc}}(t_1, t_2) = \prod_{i \neq c} \left[ (1 - f) + f e^{it_1 w_i'} \right] \cdot \left[ (1 - f) + f e^{i(t_2 + t_1 w_c')} \right].$$ (73)

Using Lemma B.2, for small $w_c'$, we apply the Taylor expansion to approximate each term:

$$(1 - f) + f e^{it_1 w_i'} \approx 1 + f(it_1 w_i'),$$ (74)

$$(1 - f) + f e^{i(t_2 + t_1 w_c')} \approx (1 - f) + f e^{it_2}.$$ (75)

Substituting back:

$$\phi_{M_{tn}, X_{tnc}}(t_1, t_2) \approx \prod_{i \neq c} (1 + fit_1 w'_i) \cdot \left[ (1 - f) + f e^{it_2} \right]. \tag{76}$$

Using Equation 59, Equation 72 and Taylor expansion, the product of the characteristic functions for the two distributions is:

$$
\begin{aligned}
\phi_{X_{tnc}}(t_2)\phi_{M_{tn}}(t_1) &= (1 - f + f e^{it_2}) \prod_{i=1}^{C} (1 - f + f e^{it_1 w'_i}) \\
&= (1 - f + f e^{it_2}) \prod_{i=1}^{C} (1 + fit_1 w'_i) \\
&= (1 - f + f e^{it_2})(1 + fit_1 w'_c) \prod_{i \neq c} (1 + fit_1 w'_i) \\
&= (1 - f + f e^{it_2}) \prod_{i \neq c} (1 + fit_1 w'_i) \\
&= \phi_{M_{tn}, X_{tnc}}(t_1, t_2)
\end{aligned} \tag{77}
$$

Thus, the joint characteristic function factorizes into the product of the marginal characteristic functions, which demonstrates that $M_{tn}$ and $X_{tnc}$ are asymptotically independent as $C \to \infty$. $\qquad\square$

**Proposition B.4.** *If $b \approx 0$, $a \geq 1$, and the firing rate $f$ is relatively small value, the PM output $Y_{tnc}$ satisfies:*

$$\mathbb{E}(Y_{tnc}) \approx \sqrt{\frac{f(1-f)}{2\pi}} \, \mathbb{E}(X_{tnc}) \tag{78}$$

$$\mathrm{Var}(Y_{tnc}) \approx \frac{f(\pi - f)}{2\pi} \, \mathrm{Var}(X_{tnc}) \tag{79}$$

*Proof.* For convenience, we denote:

$$\mu = \sum_{c=1}^{C} w'_c f, \quad \sigma^2 = \sum_{c=1}^{C} w'^2_c f(1-f) = f(1-f), \quad M'_{tn} = \mathrm{clamp}(M_{tn}, b, a). \tag{80}$$

According to Lemma B.2, the input distribution satisfies:

$$M_{tn} \sim \mathcal{N}(\mu, \sigma^2). \tag{81}$$

The expectation of the clamped variable $M'_{tnc}$ is:

$$
\begin{aligned}
\mathbb{E}(M'_{tn}) &= \int_{-\infty}^{\infty} x f(x) dx \\
&= \frac{1}{\sqrt{2\pi\sigma^2}} \int_{0}^{a} x \exp\left(-\frac{(x-\mu)^2}{2\sigma^2}\right) dx + \frac{a}{\sqrt{2\pi\sigma^2}} \int_{a}^{\infty} \exp\left(-\frac{(x-\mu)^2}{2\sigma^2}\right) dx.
\end{aligned} \tag{82}
$$

For the first term, let $t = (x - \mu)^2$, if $\mu \approx 0$, then:

$$
\begin{aligned}
&\frac{1}{\sqrt{2\pi\sigma^2}} \int_{0}^{a} x \exp\left(-\frac{(x-\mu)^2}{2\sigma^2}\right) dx \\
&= \frac{1}{2\sqrt{2\pi\sigma^2}} \int_{\mu^2}^{(a-\mu)^2} \exp\left(-\frac{t}{2\sigma^2}\right) dt + \frac{\mu}{\sqrt{2\pi\sigma^2}} \int_{0}^{a} \exp\left(-\frac{(x-\mu)^2}{2\sigma^2}\right) dx \\
&= \frac{-\sigma}{\sqrt{2\pi}} \left[ \exp\left(-\frac{t}{2\sigma^2}\right) \right]_{\mu^2}^{(a-\mu)^2} + \mu\left( \Phi\left(\frac{a-\mu}{\sigma}\right) - \Phi\left(\frac{-\mu}{\sigma}\right) \right) \\
&\approx \frac{\sigma}{\sqrt{2\pi}} \left(1 - \exp\left(-\frac{a^2}{2\sigma^2}\right)\right).
\end{aligned} \tag{83}
$$

where $\Phi(x)$ is the CDF of the standard normal distribution. The second term in the expectation is straightforward:

$$\frac{a}{\sqrt{2\pi\sigma^2}} \int_a^\infty \exp\left(-\frac{(x-\mu)^2}{2\sigma^2}\right) dx = \frac{a}{\sqrt{2\pi\sigma^2}} \int_{a-\mu}^\infty \exp\left(-\frac{t^2}{2\sigma^2}\right) dt, \tag{84}$$

Using the cumulative distribution function (CDF) again:

$$\frac{a}{\sqrt{2\pi\sigma^2}} \int_{a-\mu}^\infty \exp\left(-\frac{t^2}{2\sigma^2}\right) dt = a\left(1 - \Phi\left(\frac{a-\mu}{\sigma}\right)\right)$$
$$\approx a\left(1 - \Phi\left(\frac{a}{\sigma}\right)\right) \tag{85}$$

The $\Phi(\frac{a}{\sigma})$ and $\exp(-\frac{a^2}{2\sigma^2})$ function decay rapidly as $\sigma$ decreases. Now, combining the results from the two integrals, we have:

$$\mathbb{E}(M'_{tn}) = \frac{\sigma}{\sqrt{2\pi}} - \frac{\sigma}{\sqrt{2\pi}} \exp\left(-\frac{a^2}{2\sigma^2}\right) + a\left(1 - \Phi\left(\frac{a-\mu}{\sigma}\right)\right)$$
$$\approx \frac{\sigma}{\sqrt{2\pi}} \tag{86}$$

Based on B.3, we calculate the expectation and variance of $M'^2_{tn}$:

$$\mathbb{E}(M'^2_{tn}) = \int_{-\infty}^\infty x^2 f(x)dx$$
$$= \frac{1}{\sqrt{2\pi\sigma^2}} \int_0^a x^2 \exp\left(-\frac{x^2}{2\sigma^2}\right) dx + a^2 \cdot \int_a^\infty f(x)dx. \tag{87}$$

We calculate the first term using integration by parts. Let:

$$u = x, \quad dv = x \exp\left(-\frac{x^2}{2\sigma^2}\right) dx, \quad du = dx, \quad v = -\sigma^2 \exp\left(-\frac{x^2}{2\sigma^2}\right). \tag{88}$$

Then:

$$\frac{1}{\sqrt{2\pi\sigma^2}} \int_0^a x^2 \exp\left(-\frac{x^2}{2\sigma^2}\right) dx$$
$$= \frac{1}{\sqrt{2\pi\sigma^2}} \left(\left[-\sigma^2 x \exp\left(-\frac{x^2}{2\sigma^2}\right)\right]_0^a + \sigma^2 \int_0^a \exp\left(-\frac{x^2}{2\sigma^2}\right) dx\right)$$
$$= \frac{1}{\sqrt{2\pi\sigma^2}} \left(-\sigma^2 a \exp\left(-\frac{a^2}{2\sigma^2}\right) + \sigma^2 \int_0^a \exp\left(-\frac{x^2}{2\sigma^2}\right) dx\right). \tag{89}$$

The remaining integral is a standard normal distribution integral:

$$\frac{\sigma^2}{\sqrt{2\pi\sigma^2}} \int_0^a \exp\left(-\frac{x^2}{2\sigma^2}\right) dx = \sigma^2 \left(\Phi\left(\frac{a}{\sigma}\right) - \frac{1}{2}\right), \tag{90}$$

where $\Phi(x)$ is the CDF of the standard normal distribution.

Substituting (90) into (89):

$$\frac{1}{\sqrt{2\pi\sigma^2}} \int_0^a x^2 \exp\left(-\frac{x^2}{2\sigma^2}\right) dx = \frac{-a\sigma}{\sqrt{2\pi}} \exp\left(-\frac{a^2}{2\sigma^2}\right) + \sigma^2 \left(\Phi\left(\frac{a}{\sigma}\right) - \frac{1}{2}\right). \tag{91}$$

The second term is the tail of the normal distribution:

$$\int_a^\infty f(x)dx = \Phi\left(-\frac{a}{\sigma}\right), \tag{92}$$

we have:

$$a^2 \cdot \int_a^\infty f(x)dx = a^2 \Phi\left(-\frac{a}{\sigma}\right). \tag{93}$$

Combining (91) and (93) into (87), we get:

$$\mathbb{E}(M_{tn}'^2) = \frac{-a\sigma}{\sqrt{2\pi}}\exp\left(-\frac{a^2}{2\sigma^2}\right) + \sigma^2\left(\Phi\left(\frac{a}{\sigma}\right) - \frac{1}{2}\right) + a^2\Phi\left(-\frac{a}{\sigma}\right)$$
$$\approx \frac{\sigma^2}{2}. \tag{94}$$

Since $\Phi\left(-\frac{a}{\sigma}\right)$ is exponentially small for moderate $a$, the term $a^2\Phi\left(-\frac{a}{\sigma}\right)$ is negligible compared to leading terms and is often omitted for simplicity.

Using $\mathrm{Var}(M_{tn}') = \mathbb{E}(M_{tn}'^2) - \mathbb{E}(M_{tn}')^2$, we calculate:

$$\mathrm{Var}(M_{tn}') \approx \frac{\sigma^2}{2} - \left[\frac{\sigma}{\sqrt{2\pi}}\left(1 - \exp\left(-\frac{a^2}{2\sigma^2}\right)\right)\right]^2$$
$$\approx \frac{\pi - 1}{2\pi}\sigma^2$$
$$= \frac{\pi - 1}{2\pi}f(1-f). \tag{95}$$

Given that $Y_{tnc} = M_{tnc}' \cdot X_{tnc}$, and based on Lemma B.3 that the distributions of $X_{tnc}$ and $M_{tn}'$ can be considered independent, the expectation of $Y_{tnc}$ is:

$$\mathbb{E}(Y_{tnc}) = \mathbb{E}(M_{tn}') \cdot \mathbb{E}(X_{tnc})$$
$$\approx \sqrt{\frac{f(1-f)}{2\pi}}\mathbb{E}(X_{tnc}). \tag{96}$$

The variance of $Y_{tnc}$ is computed as:

$$\mathrm{Var}(Y_{tnc}) = \mathrm{Var}(M_{tn}') \cdot \mathrm{Var}(X_{tnc}) + \mathrm{Var}(M_{tn}') \cdot \mathbb{E}(X_{tnc})^2 + \mathrm{Var}(X_{tnc}) \cdot \mathbb{E}[M_{tn}']^2$$
$$= \frac{f(\pi - f)}{2\pi}f(1-f)$$
$$\approx \frac{f(\pi - f)}{2\pi}\mathrm{Var}(X_{tnc}). \tag{97}$$

Thus, the proposition is proven:

$$\mathbb{E}(Y_{tnc}) \approx \sqrt{\frac{f(1-f)}{2\pi}}\mathbb{E}(X_{tnc}), \quad \mathrm{Var}(Y_{tnc}) \approx \frac{f(\pi - f)}{2\pi}\mathrm{Var}(X_{tnc}). \tag{98}$$

$\square$

In practice, we recommend setting the hyperparameters as follows: $b = 0$ and $a \in [1, 2]$. Setting $b = 0$ allows the processing module to completely eliminate certain features in the spatial domain. Furthermore, selecting $a \in [1, 2]$ enables the processing module to selectively enhance specific spatial features. This also ensures that both the mean and variance do not become too large or too small, maintaining the numerical stability.

## B.3 Gradient Analysis

This section on the derivation of the traditional SNN network is mainly referenced from [40, 7, 8]. First, we derive the temporal gradient of the traditional SNN network, where the temporal gradient is primarily backpropagated through the membrane potential. Taking the vanilla LIF neuron as an example, we use the following form to analyze the gradient problem:

$$\mathbf{H}^l(t+1) = \left(1 - \frac{1}{\tau}\right)\left(\mathbf{H}^l(t) - \vartheta\mathbf{S}^l(t)\right) + \mathbf{W}^l\mathbf{S}^{l-1}(t+1), \tag{99}$$

The derivative of the loss with respect to the weights $W_l$ is:

$$\nabla_{\mathbf{W}^l}\mathcal{L} = \sum_{t=0}^{T-1}\frac{\partial\mathcal{L}}{\partial\mathbf{H}^l(t)}^\top\mathbf{S}^{l-1}[t]^\top, l = L, L-1, \cdots, 1, \tag{100}$$

The gradient expression can be written as:

$$\frac{\partial \mathcal{L}}{\partial \mathbf{H}^l(t)} = \underbrace{\frac{\partial \mathcal{L}}{\partial \mathbf{H}^{l+1}(t)} \frac{\partial \mathbf{H}^{l+1}(t)}{\partial \mathbf{S}^l(t)} \frac{\partial \mathbf{S}^l(t)}{\partial \mathbf{H}^l(t)}}_{Spatial\ Gradient} +$$

$$\underbrace{\sum_{t'=t+1}^{T-1} \frac{\partial \mathcal{L}}{\partial \mathbf{H}^{l+1}(t')} \frac{\partial \mathbf{H}^{l+1}(t')}{\partial \mathbf{S}^l(t')} \frac{\partial \mathbf{S}^l(t')}{\partial \mathbf{H}^l(t')} \prod_{t''=1}^{t'-t} \epsilon^L(t'-t''), l < L,}_{Temporal\ Gradient} \tag{101}$$

$$\frac{\partial \mathcal{L}}{\partial \mathbf{H}^l(t)} = \underbrace{\frac{\partial \mathcal{L}}{\partial \mathbf{S}^l(t)} \frac{\partial \mathbf{S}^l(t)}{\partial \mathbf{H}^l(t)}}_{Spatial\ Gradient} + \underbrace{\sum_{t'=t+1}^{T-1} \frac{\partial \mathcal{L}}{\partial \mathbf{S}^l(t')} \frac{\partial \mathbf{S}^l(t')}{\partial \mathbf{H}^l(t')} \prod_{t''=1}^{t'-t} \epsilon^L(t'-t''), l = L,}_{Temporal\ Gradient} \tag{102}$$

$\epsilon^L$ is defined as the sensitivity of the membrane potential $H^l(t+1)$ with respect to $H^l(t)$ between adjacent timesteps.

$$\epsilon^l(t) \equiv \frac{\partial \mathbf{H}^l(t+1)}{\partial \mathbf{H}^l(t)} + \frac{\partial \mathbf{H}^l(t+1)}{\partial \mathbf{S}^l(t)} \frac{\partial \mathbf{S}^l(t)}{\partial \mathbf{H}^l(t)}. \tag{103}$$

If we use a simple rectangular function as a surrogate for the gradient.

$$\epsilon^l(t)_{jj} = \begin{cases} 0, & \frac{1}{2}\vartheta < H_j^l(t) < \frac{3}{2}\vartheta, \\ 1 - \frac{1}{\tau}, & \text{otherwise}. \end{cases} \tag{104}$$

From the above equation, it can be concluded that if the membrane potential approaches the threshold at any given timestep, the temporal gradient $\prod_{t''=1}^{t'-t} \epsilon^L(t'-t'')$ will vanish. This highlights a common issue with temporal gradients in the vanilla LIF model, which remains a problem even with short timesteps.

Next, we perform gradient analysis on neurons with a feedback structure. Assume the structure of the feedback is $\varphi$, which includes PM and CM.

$$\mathbf{H}^l(t+1) = \left(1 - \frac{1}{\tau}\right)\left(\mathbf{H}^l(t) - \vartheta \mathbf{S}^l(t)\right) + \mathbf{W}^l \mathbf{S}^{l-1}(t+1) + \varphi_\theta(\mathbf{S}^l(t)) \tag{105}$$

Following the above derivation, we similarly define the variable $\epsilon$:

$$\epsilon^l(t) \equiv \frac{\partial \mathbf{H}^l(t+1)}{\partial \mathbf{H}^l(t)} + \frac{\partial \mathbf{H}^l(t+1)}{\partial \mathbf{S}^l(t)} \frac{\partial \mathbf{S}^l(t)}{\partial \mathbf{H}^l(t)} + \underbrace{\frac{\partial \mathbf{H}^l(t+1)}{\partial \varphi_\theta(\mathbf{S}^l(t))} \frac{\partial \varphi_\theta(\mathbf{S}^l(t))}{\partial \mathbf{S}^l(t)} \frac{\partial \mathbf{S}^l(t)}{\partial \mathbf{H}^l(t)}}_{Feedback\ gradient} \tag{106}$$

$$\epsilon^l(t) = \left(1 - \frac{1}{\tau}\right) - \left(1 - \frac{1}{\tau}\right)\vartheta \cdot \frac{\partial \mathbf{S}^l(t)}{\partial \mathbf{H}^l(t)} + \frac{\partial \varphi_\theta(\mathbf{S}^l(t))}{\partial \mathbf{S}^l(t)} \frac{\partial \mathbf{S}^l(t)}{\partial \mathbf{H}^l(t)} \tag{107}$$

Similarly we have:

$$\epsilon^l(t)_{jj} = \begin{cases} \frac{\partial \varphi_\theta(\mathbf{S}^l(t))}{\partial \mathbf{S}^l(t)}, & \frac{1}{2}\vartheta < H_j^l(t) < \frac{3}{2}\vartheta, \\ 1 - \frac{1}{\tau}, & \text{otherwise}. \end{cases} \tag{108}$$

Then, in training, $\frac{\partial \varphi_\theta(\mathbf{S}^l(t))}{\partial \mathbf{S}^l(t)}$ is not possible to be zero.

## C   Supplementary Results

### C.1   Energy Consumption Calculation of TDFormer

This section is mainly referenced from [3]. We calculate the number of Synaptic Operations (SOPs) of spike before calculating theoretical energy consumption for TDFormer.

$$\text{SOP} = f_r \times T \times \text{FLOPs} \tag{109}$$

Table 5: Results averaged across seeds: 0, 42, 2024, 3407 and 114514. Bold results indicate superior performance compared to the baselines.

| Methods | Dataset/Time Step | Architecture | Baseline | CM1+V1 |
|---------|-------------------|--------------|----------|--------|
| SpikeformerV1 | CIFAR-10/T = 2 | Spikformer-2-384 | **94.18±0.06** | 94.07±0.07 |
| | CIFAR-10/T = 4 | | 94.84±0.14 | **94.86±0.05** |
| | CIFAR-10/T = 2 | Spikformer-4-384 | 93.65±0.23 | **94.05±0.14** |
| | CIFAR-100/T =2 | | 75.25±0.19 | **75.99±0.12** |
| | CIFAR-10/T = 4 | | 94.73±0.06 | **95.13±0.07** |
| | CIFAR-100/T = 4 | | 77.56±0.22 | **77.60±0.26** |
| | CIFAR-10/T = 6 | | 95.09±0.08 | **95.16±0.14** |
| | CIFAR-100/T = 6 | | **78.21±0.22** | 77.99±0.05 |
| | CIFAR10-DVS/T = 10 | | 78.08±0.70 | **78.13±0.72** |
| | CIFAR10-DVS/T= 16 | | 79.40±0.36 | **80.20±0.75** |
| SDTV1 | CIFAR-10/T = 4 | Spiking Transformer-2-512 | 95.76±0.06 | **95.92±0.02** |
| | CIFAR-100/T =4 | | 79.15±0.14 | **79.35±0.16** |
| | CIFAR-10/T = 4 | Spiking Transformer-2-256 | 94.47±0.11 | **94.64±0.04** |
| | CIFAR-100/T =4 | | 76.15±0.13 | **76.26±0.13** |
| | DVS128 Gesture/T=10 | | 96.79±0.67 | **96.92±0.29** |
| | DVS128 Gesture/T=16 | | 97.98±0.59 | **99.04±0.28** |
| | CIFAR10-DVS/T = 10 | | 75.03±0.67 | **75.05±0.11** |
| | CIFAR10-DVS/T = 16 | | 77.07±0.19 | **77.45±0.43** |

where $f_r$ is the firing rate of the block and $T$ is the simulation time step of spiking neuron. FLOPS refers to floating point operations of block, which is the number of multiply-and-accumulate (MAC) operations and SOP is the number of spike-based accumulate (AC) operations.

$$E_{\text{TDFormer}} = E_{Baseline} + E_{AC} \times (\text{SOP}_{\text{PM}} + \text{SOP}_{\text{CM}}) \tag{110}$$

The channel-wise token mixer in TDFormer is highly power-efficient, consisting of only a linear layer, a LIF neuron, and a BN layer. The BN parameters can be fused into the linear layer via reparameterization, making its power consumption negligible. The linear layer maintains a constant channel dimension, resulting in much lower power usage than conventional MLPs. Furthermore, the spatial-wsie token mixer in PM has a time complexity of only $O(ND)$, which is much lower than the $O(N^2D)$ of SSA. In the CM module, although a token mixer is used, the firing rates in both PM and CM are very low. In our experiments, we observed that the firing rate in both modules remains around 0.05. As a result, the overall power overhead of TDFormer is marginal.

## C.2 Additional Experiments and Visualizations

Table 6: Results of different TDFormer variants. The results in bold indicate superior performance compared to the baseline. The default configuration used in our work is indicated by *. CM1-CM3 denote different strategies for integrating top-down information with bottom-up features. CM1: $S_{td}$ is fused into the computation of the attention map. CM2: $S_{td}$ is fused into the value of self-attention. CM3: $S_{td}$ is incorporated into the input of the attention module.

| Model Type | SpikeformerV1 (Spikformer-4-384) | | | SDTV1 (Spiking Transformer-2-256) | | |
|---|---|---|---|---|---|---|
| | Acc (%) | FLOPs (G) | Param (M) | Acc (%) | FLOPs (G) | Param (M) |
| Baseline | 94.73 | 3.71 | 9.33 | 94.47 | 1.25 | 2.57 |
| *CM1+V1 | **95.14** | 3.88 | 9.92 | **94.77** | 1.31 | 2.69 |
| CM1+V2 | **94.79** | 3.88 | 9.92 | **94.93** | 1.31 | 2.69 |
| CM1+V3 | **94.90** | 3,88 | 9.92 | **94.61** | 1.31 | 2.69 |
| CM1+V4 | **94.94** | 3.88 | 9.92 | **94.88** | 1.31 | 2.69 |
| CM2+V1 | **94.88** | 3.88 | 9.92 | **94.73** | 1.31 | 2.69 |
| CM2+V2 | **94.75** | 3.88 | 9.92 | **94.79** | 1.31 | 2.69 |
| CM2+V3 | 94.70 | 3.88 | 9.92 | **94.75** | 1.31 | 2.69 |
| CM2+V4 | **95.27** | 3.88 | 9.92 | **94.66** | 1.31 | 2.69 |
| CM3+V1 | 94.69 | 3.90 | 9.92 | 94.43 | 1.32 | 2.69 |
| CM3+V2 | **94.89** | 3.90 | 9.92 | **94.69** | 1.32 | 2.69 |
| CM3+V3 | 94.35 | 3.90 | 9.92 | 93.94 | 1.32 | 2.69 |
| CM3+V4 | **94.90** | 3.90 | 9.92 | **94.61** | 1.32 | 2.69 |

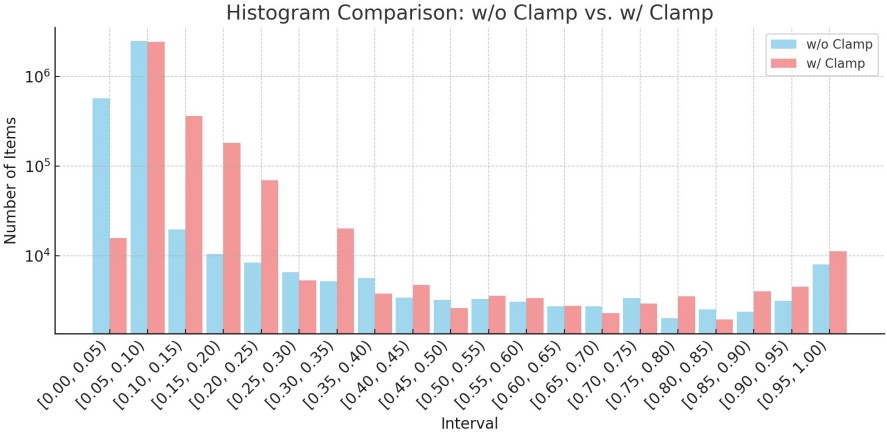

Figure 3: This is the histogram of the gradient of the surrogate function for LIF neurons in the attention module within the PM model. From the figure, we can see that the clamp operation ensures that the variance in the attention map does not become too large, thus preventing the vanishing gradient problem.

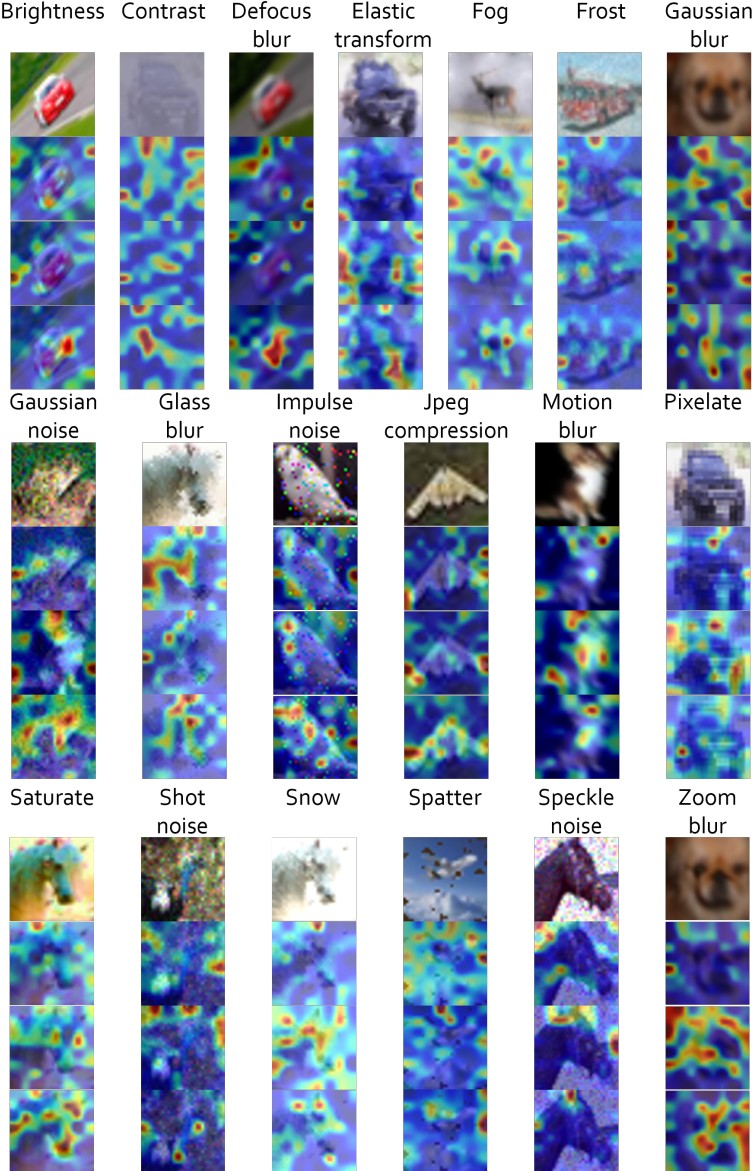

Figure 4: Visualization of CIFAR-10C. This figure showcases 19 columns corresponding to 19 different types of corruptions. Each column contains four images: the top image displays the original CIFAR-10C image; the second image shows the visualization result of the baseline model; the third image illustrates the first feedforward stage of the TDFormer model; the fourth image depicts the second feedforward stage of the TDFormer model, demonstrating the model's dynamic attention adjustments across stages.

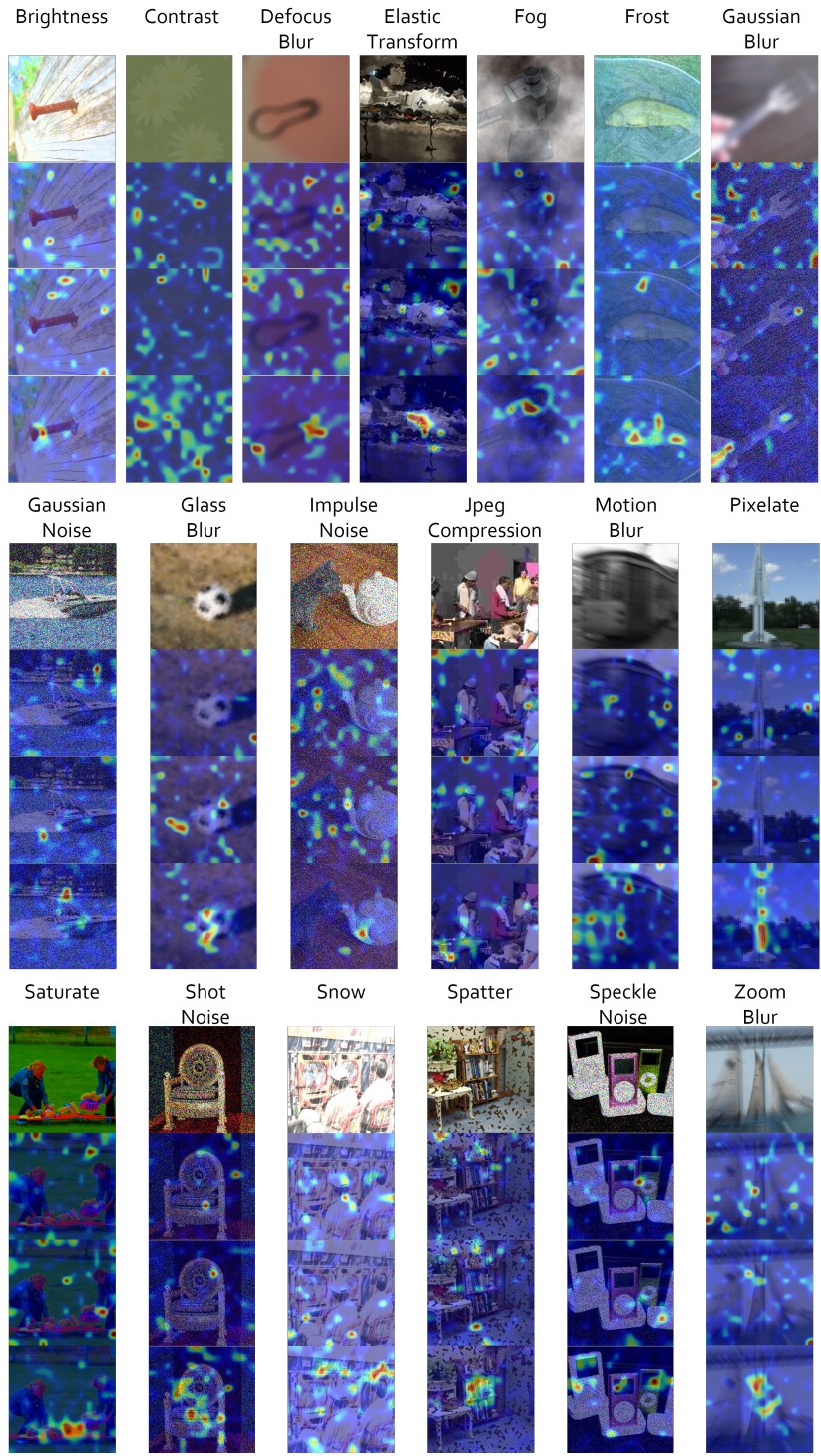

Figure 5: Visualization of ImageNet-C. This figure showcases 19 columns corresponding to 19 different types of corruptions. The layout and visualization style are similar to those shown in Figure 4.

Table 7: Robustness comparison on the CIFAR-10C dataset. The results in bold indicate superior performance compared to the baseline. Average performance across different distortion types is indicated by *.

| Corruption Type | SpikformerV1 /TDFormer | Time Step | SpikformerV1 /TDFormer | Corruption Type |
|---|---|---|---|---|
| | Acc (%) | | Acc (%) | |
| Brightness | 91.32/91.27 (-0.05) | 1 | **76.23/76.97 (+0.74)** | Motion Blur |
| | **91.87/91.94 (+0.06)** | 2 | **77.00/78.30 (+1.30)** | |
| | **93.14/93.29 (+0.15)** | 4 | **79.44/80.01 (+0.57)** | |
| Contrast | **69.93/70.40 (+0.47)** | 1 | **79.31/79.51 (+0.20)** | Pixelate |
| | **70.41/71.25 (+0.84)** | 2 | 78.70/78.67 (-0.03) | |
| | 77.06/76.57 (-0.49) | 4 | **81.14/81.45 (+0.31)** | |
| Defocus Blur | **80.59/80.83 (+0.24)** | 1 | 87.33/87.10 (-0.23) | Saturate |
| | **81.39/82.15 (+0.76)** | 2 | **88.30/88.44 (+0.14)** | |
| | 82.88/82.75 (-0.13) | 4 | **90.58/90.60 (+0.02)** | |
| Elastic Transform | **84.00/84.05 (+0.05)** | 1 | **69.63/70.68 (+1.05)** | Shot Noise |
| | **84.10/84.63 (+0.53)** | 2 | **70.96/71.09 (+0.13)** | |
| | 85.54/85.52 (-0.02) | 4 | **73.23/73.32 (+0.09)** | |
| Fog | **84.29/85.22 (+0.93)** | 1 | **84.47/84.71 (+0.24)** | Snow |
| | **85.09/85.75 (+0.66)** | 2 | **84.72/84.72 (+0.00)** | |
| | **87.25/87.53 (+0.28)** | 4 | **86.90/87.18 (+0.28)** | |
| Frost | **82.35/82.66 (+0.31)** | 1 | 88.20/88.03 (-0.17) | Spatter |
| | **83.04/83.27 (+0.23)** | 2 | **87.58/87.71 (+0.13)** | |
| | **85.46/85.70 (+0.24)** | 4 | 89.14/89.02 (-0.12) | |
| Gaussian Blur | **73.33/74.05 (+0.72)** | 1 | **71.77/72.66 (+0.89)** | Speckle Noise |
| | **74.79/75.84 (+1.05)** | 2 | 72.66/72.64 (-0.02) | |
| | **76.08/76.25 (+0.17)** | 4 | **75.10/75.37 (+0.27)** | |
| Gaussian Noise | **61.35/62.71 (+1.36)** | 1 | **75.98/76.68 (+0.70)** | Zoom Blur |
| | 63.05/62.71 (-0.34) | 2 | **77.60/78.75 (+1.15)** | |
| | **64.34/64.89 (+0.55)** | 4 | **78.68/79.14 (+0.46)** | |
| Impulse Noise | **67.84/68.10 (+0.26)** | 1 | **57.86/58.26 (+0.40)** | Glass Blur |
| | 65.83/65.36 (-0.47) | 2 | 56.09/55.81 (-0.28) | |
| | **65.98/66.93 (+0.95)** | 4 | **59.43/60.46 (+1.03)** | |
| JPEG Compression | **83.32/83.53 (+0.21)** | 1 | **78.11/78.55 (+0.44)** | * Avg |
| | **83.93/84.00 (+0.07)** | 2 | **78.52/78.84 (+0.32)** | |
| | **84.60/84.76 (+0.16)** | 4 | **80.53/80.78 (+0.25)** | |

# D Limitations, Future Work, and Broader Impacts

## D.1 Limitations

Despite the promising enhancements introduced by our proposed TDFormer with top-down feedback structure for spiking neural networks, several limitations remain. First, the current feedback mechanism is specifically designed for Transformer-based architectures and may not be directly applicable to CNN-based SNNs, limiting its architectural generalizability. Second, our evaluation has so far been limited to image classification tasks, which may not fully reflect the method's effectiveness in other domains such as object detection[42], semantic segmentation[43], and NLP tasks[44].

## D.2 Future Work

Future work could focus on generalizing the proposed TDFormer architecture to other network backbones, such as CNN-based spiking neural networks, thereby improving its architectural compatibility

and deployment flexibility. In addition, extending the evaluation of TDFormer to tasks such as object detection, semantic segmentation, and natural language processing would provide deeper insights into its generalization capacity across diverse domains and data modalities. Moreover, we observe that the proposed top-down feedback structure increases the diversity of spike patterns[10], which may contribute to the observed performance gains. Investigating the underlying relationship between spike diversity and task performance remains an important direction for future research.

### D.3 Broader Impacts

This paper focuses on the fundamental research of spiking neural networks, introducing a top-down feedback structure that aims to enhance their performance. Generally, there are no negative societal impacts in this work.

