# OpenReview forum: "TDFormer: Top-Down Attention-Controlled Spiking Transformer"
_NeurIPS.cc/2025/Conference — Submitted to NeurIPS 2025_

### Official Review · Reviewer_Sx25 · 2025-06-01

**Clarity:** 2
**Significance:** 2
**Originality:** 2
**Rating:** 1
**Confidence:** 5

**Summary:**

This article introduces a Top-Down Attention-Controlled Spiking Transformer (TDFormer), which aims to improve the performance of SNNs by incorporating top-down attention mechanisms inspired by human brain processes.

**Questions:**

1. While the proposed method demonstrates promising results, it would be more convincing to evaluate its performance on a more advanced architecture such as Spikformer v2 [1], which may better reveal its generalizability and scalability. Additionally, the method presents a trade-off between power consumption and accuracy—raising the question of whether further optimization could help reduce power usage without significantly compromising performance. Exploring such improvements would enhance the practical applicability of the method, especially for energy-constrained scenarios.


[1] Spikformer v2: Join the high accuracy club on imagenet with an snn ticket

**Ethical Concerns:**

["NO or VERY MINOR ethics concerns only"]

**Final Justification:**

The proposed TDFormer doesn't achieve some training speed improvements, and the performance gains are marginal and may fall within the margin of reproduction error. Given the limited accuracy improvement and the added power overhead, we believe the overall contribution is not sufficiently strong. Despite the acceleration benefits, the paper lacks compelling empirical advances and does not meet the bar for acceptance.

**Limitations:**

1. The proposed method has a trade-off between power consumption and accuracy, meaning that achieving higher accuracy may come at the cost of increased energy usage, while reducing power consumption could lead to a slight degradation in performance.

2. The experimental results appear to be highly dependent on the strength of the baseline. When evaluated against more competitive baselines, the proposed method shows only limited improvements, suggesting that its overall effectiveness may be modest.

**Quality:**

2

**Strengths And Weaknesses:**

The paper presents a well-written and clearly articulated study, offering a method that effectively integrates principles from neuroscience with SNNs.

This interdisciplinary approach enhances the biological plausibility of the model and contributes meaningful insights to the development of energy-efficient neural computation.

---

> ### Author Rebuttal · Authors · 2025-07-31
>
> We sincerely thank the reviewer for the detailed and constructive feedback. We address each of the raised concerns below:
>
> ---
>
> **Question 1: Evaluation on more advanced models like Spikformer V2**
>
> **Answer:**
> We appreciate the reviewer’s suggestion regarding the use of more advanced models
> for evaluation. However, we would like to clarify a misunderstanding:
>
> 1. **Qkformer was the most advanced model available at the time of our submission (excluding Spike-driven V3)**. QKFormer significantly outperforms Spikformer V2 in top-1 accuracy on ImageNet at 224 resolution (e.g., 84.22% vs. 80.38%), as reported in the original paper. The reason we did not evaluate on Spike-driven V3 is that its integer-based training removes the temporal dimension from the network dynamics, making our method inapplicable.
>
> 2. **Qkformer is also a more recent and peer-reviewed work**, published in October
> 2024, whereas Spikformer V2 remains a preprint uploaded in January 2024.
>
> Therefore, we believe that the significant improvements demonstrated on Qkformer, one of the most advanced models at the time, across multiple datasets, especially achieving a state-of-the-art accuracy of 86.83% on ImageNet, provide strong evidence of the effectiveness of our method.
>
> ---
>
> **Question 2: Concern about the method’s applicability under
> resource constraints due to the power-accuracy trade-off**
>
> **Answer:**
> While our method slightly increases the number of parameters and power
> consumption, it achieves a significantly better **performance-to-power ratio**,
> which demonstrates its effectiveness for energy-constrained scenarios.
>
> 1. In **Table 1**, our lightweight variant increases power consumption from 38.91
> to 38.93 (less than a **1%** increase), and parameters from 64.96M to 65.55M
> (a **0.91%** increase). However, accuracy improves from 84.22% to 85.37%,
> even surpassing Qkformer at 288 resolution (85.37% vs. 85.20%). In addition, our
> power usage is still significantly lower than Qkformer (39.93mJ vs. 64.39mJ),
> representing a **38.0%** reduction.
>
> 2. At 224 resolution, our model achieves 85.57% accuracy, which is comparable to
> Qkformer at 384 resolution (85.57% vs. 85.65%), while consuming much less power
> (39.10mJ vs. 113.64mJ), resulting in a **65.6%** reduction in energy cost.
>
>
> In addition, we conducted ablation experiments based on Spikformer to further
> validate that the performance improvement comes from our proposed method rather
> than from a trade-off between accuracy and increased power or parameters.
>
>
> When applying our method to **Spikformer-2-384**, we achieved **better performance than the baseline Spikformer-4-384** (95.02% vs. 94.73%) while **reducing parameters by 33%** (6.21M vs. 9.33M).
> In addition, Spikformer-3-384 (TDAC) further improved performance to **95.14%**, with a **14.36%** reduction in parameters (7.99M vs. 9.33M).
>
> These results clearly demonstrate that our method achieves **a better
> performance-to-energy ratio**, making it highly suitable for energy-constrained
> deployment scenarios. All results are averaged over five
> random seeds for fairness.
>
> | Model                    | Timestep | Parameters | Accuracy       |
> |--------------------------|----------|------------|----------------|
> | Spikformer-4-384         |     4        | 9.33M      | 94.73 ± 0.06   |
> | Spikformer-2-384 (TDAC)  |     4        | 6.21M      | **95.02 ± 0.068**   |
> | Spikformer-3-384 (TDAC)  |   4        | 7.99M      | **95.14 ± 0.013**  |
>
> **Table 4: TDAC achieves higher accuracy with fewer parameters on Spikformer**
>
> ---
>
> **Question 3: Limited improvement over competitive baselines**.
>
> **Answer:**
>  When applying our method to Qkformer, as shown in **Table 1**, it achieves a new
> state-of-the-art accuracy of **86.83%** using only **69.09M parameters** and
> **113.79 mJ** of energy. Furthermore, we consistently observe improvements of over 1% on **ImageNet** across multiple input resolutions. On **CIFAR10-DVS**, our method achieves a significant gain of
> **1.83%** at 16 simulation steps (85.83% vs. 84.00%). These results clearly demonstrate that our gains are not "only limited
> improvements," but instead represent meaningful and practical advances over the competitive baselines.
>
> ---

---

> > ### Comment · Reviewer_Sx25 · 2025-08-01
> >
> > The performance improvement of TDFormer on CIFAR10-DVS appears marginal compared to the baseline model (and could potentially fall within the margin of reproduction error). Moreover, it incurs higher power consumption. I would also like to ask the authors: can the proposed method accelerate the training of SNNs? Since SNN training typically has a complexity of $\mathcal{O}(T)$, what is the actual training time of the largest TDFormer variant based on QKFormer on ImageNet? Have the authors achieved any acceleration in this regard?

---

> > > ### Comment · Reviewer_Sx25 · 2025-08-02
> > >
> > > Since the author did not address the questions I raised, I found it difficult to fully evaluate the work and therefore decided to lower my score.

---

> ### Author Response · Authors · 2025-08-05
> **Additional Experiments Addressing Reviewer Concerns**
>
> During the **discussion period**, we conducted additional experiments to address your concerns, as summarized below:
>
> ---
>
> **1. Marginal Gains on CIFAR10-DVS**
>
> To address your concern regarding the **marginal improvement** on the CIFAR10-DVS dataset, we conducted further experiments. The results are as follows:
>
>
> | Model            | Timestep | Dataset | Acc(%) |
> |------------------|----------|----------|-----------------|
> | Spikformer | 10        |   CIFAR10-DVS |    78.08*     |
> | Spikformer | 16        |      CIFAR10-DVS |     79.40*       |
> | Qkformer | 16        |      CIFAR10-DVS |    84.0       |
> | Spikformer-TDAC | 10    |    CIFAR10-DVS |     **80.1(+2.02)**     |
> | Spikformer-TDAC  | 16    |   CIFAR10-DVS |    **82.3(+2.90)**      |
> | Qkformer-TDAC  | 16    |   CIFAR10-DVS |   **85.83(+1.83)**     |
>
> **Table 5: Additional Experiments on CIFAR10-DVS**
>
> Numbers with ∗ denote our implementation, representing results that differ from those reported in the original papers. These results demonstrate that the performance improvement of our method on CIFAR10-DVS **is not merely marginal**.
>
> ---
>
> **2. Inference time Evaluation on ImageNet**
>
> To better evaluate the **training and inference efficiency** of our method, we conducted inference-time measurements on the **largest QKFormer model (HST-10-768)**. The results are shown in the table below:
>
> | Model            | Timestep | Params | Energy| Training time |  Acc(%)     |
> |------------------|----------|------------|-------------|-------------------------------|---------|
> | QKFormer-10-768  | 4        | 64.96      | 38.91       | 8.51 s             |     84.22          |
> | TDFormer-10-768  | 4     | **65.55 (+0.91%)** | **38.93 (+0.05%)**  | **8.78 s (+3.17%)**  |       **85.37(+1.15)**     |
> **Table 6: Inference time Evaluation**
>
> The number of parameters is expressed in millions (M), energy consumption in millijoules (mJ), and training time is measured **on a single batch with a batch size of 60**.  Despite the negligible overhead in energy consumption (+0.05%), parameter count (+0.91%), and inference time (+3.17%), our model achieves a significant accuracy improvement, highlighting its superior performance–efficiency trade-off.
>
>
> ---
>
> **3. Can the proposed method help accelerate the training process of SNNs?**
>
>
> Our proposed method does not accelerate the training process of SNNs. Due to the introduction of feedback connections across timesteps, it inevitably incurs a slight increase in energy consumption and parameter count, which leads to longer training times—as shown in Table 6. This overhead is mainly observed under the standard PyTorch framework, where sequential execution limits temporal parallelism.
>
> ---
>
> **4. Are there any ways to accelerate the proposed method?**
>
> While our method introduces feedback connections across timesteps—where outputs from previous steps are used to influence subsequent computations—this does not fundamentally imply a proportional increase in training time. The current implementation, **based on PyTorch and SpikingJelly**, any form of temporal dependency inherently incurs latency, since later timesteps must wait for the completion of earlier ones. However, this is a limitation of the high-level framework's execution model, not of our method itself.
>
> In theory, **this sequential bottleneck can be overcome through a custom CUDA implementation**. For example, intermediate results from earlier timesteps can be buffered and accessed via shared memory or device-level parallelism, allowing multiple timesteps to be computed in an overlapped manner. Moreover, since our feedback mechanism is relatively lightweight—only requiring simple operations like accumulation or gating—it is well-suited to efficient hardware acceleration.  Therefore, **while the current PyTorch-based implementation does not realize this potential due to framework constraints**, the proposed method could theoretically achieve comparable training speed to non-feedback SNNs when implemented with custom low-level CUDA kernels optimized for asynchronous execution.

---

> > ### Author Response · Authors · 2025-08-09
> >
> > Dear Reviewer Sx25,
> >
> > Thank you for your valuable feedback and for taking the time to engage with our submission.
> >
> > However, we would like to clarify that **from the time you raised your concerns until you decided to lower your score**, we were actively **conducting experiments and preparing responses to address the points you raised**. Throughout this entire period, we worked diligently on the experiments and updated our responses to ensure we addressed your concerns comprehensively.
> >
> > We carefully revised our submission and submitted our responses within the official discussion period, which ended on **August 8, 11:59 PM AoE**. We respectfully believe that **lowering your score before we had the opportunity to provide our responses, especially when several days remained in the discussion phase, is not entirely fair**.
> >
> > We welcome any additional feedback regarding our method or the responses during the discussion phase, as we are committed to improving our work based on constructive comments.

---

> ### Comment · Reviewer_Sx25 · 2025-08-09
> **Reviewer Response**
>
> While I acknowledge the authors’ additional experiments and clarifications, my main concerns remain unaddressed.
>
> 1. **Marginal Gains & Reproducibility Concerns**
>    The reported accuracy improvements on CIFAR10-DVS (e.g., +1.83% for QKFormer) are small and could fall within the margin of reproduction error, particularly given the variability seen in event-based datasets. The added experiments do not convincingly demonstrate that the gains are statistically significant or robust across settings.
>
> 2. **Increased Overhead Without Speed Benefit**
>    The authors confirm that the proposed TDFormer does not accelerate SNN training; in fact, it increases training time (+3.17%) and parameter count (+0.91%), and introduces additional energy consumption, albeit small (+0.05%). In my view, the lack of training speed improvement combined with added overhead undermines the practical benefit.
>
> 3. **Lack of Novelty**
>    The core mechanism is a direct adaptation of an existing ANN approach (\[1] Shi et al., CVPR 2023) with minimal modification. This method transfer is not tailored to the specific properties of SNNs—temporal sparsity, event-driven computation—and, as such, does not fully leverage the unique advantages of the SNN domain.
>
> 4. **Domain Suitability**
>    More critically, the borrowed mechanism appears ill-suited for SNNs, as evidenced by the negligible accuracy gains (<2%) alongside increased power and training cost. This suggests a mismatch between the method’s design assumptions and the constraints/opportunities in SNN architectures.
>
> Given the limited empirical improvement, absence of speed-up, added overhead, and lack of substantive methodological innovation, I believe the overall contribution is not sufficiently strong for NeurIPS. I would recommend the authors consider a venue such as *Neural Networks*, where a more incremental adaptation of ANN ideas to SNNs might be a better fit.
>
> **Reference**
>
> \[1] Shi, Baifeng, Trevor Darrell, and Xin Wang. "Top-down visual attention from analysis by synthesis." *CVPR*, 2023.
>
> .

---

### Official Review · Reviewer_CbQg · 2025-06-09

**Clarity:** 2
**Significance:** 2
**Originality:** 2
**Rating:** 3
**Confidence:** 4

**Summary:**

This paper introduces TDFormer, which integrates top-down attention-controlled modules (TDAC) into Transformer-based SNNs, incorporating the biological system-inspired top-down attention mechanism to enhance the performance and robustness. However, the proposed Processing Module (PM) and Control Module (CM) within the TDAC closely resemble the design of prior work [1], which raises concerns about the novelty and contributions of this paper. Additionally, the two feedforward stages introduced in the paper result in increased FLOPs, memory, and parameter costs, as well as higher inference latency. These factors are not adequately reflected in the experimental comparison tables. While the proposed method yields marginal performance improvement, it significantly undermines the inherent advantages of SNNs, such as low power consumption, low computational cost, and fast inference.

**Questions:**

1. The proposed method requires two inferences, with the equivalent number of time steps being close to doubling the baseline's time steps, which also nearly doubles the energy consumption. I doubt that this method outperforms simply doubling the time steps of the baseline.

2. The biological relevance of the paper is questionable. Does TDFormer truly simulate the attention mechanisms in the human brain? Although the paper mentions the existence of top-down attention mechanisms in biological systems, is this simulation accurate enough? Are there other mechanisms or model architectures that can more closely mimic the attention mechanisms of the human brain? In traditional autoregressive attention transformer structures, newly generated words are fed back into the input to guide the next generation.

3. The inference and training costs are close to double that of the original structure. This paper does not provide pseudocode for the training process. During training, will spiking attention use BPTT to propose a decline? Additionally, why does using both L1 and L2 in the loss function promote model convergence? The paper does not provide a reasonable description.

4. This paper lacks visual case studies to show what changes occur in the attention maps, output attention features, or membrane potentials after top-down guidance. Without visualization experiments, it is difficult to convince that top-down attention has the advantage.

[1] Shi, Baifeng, Trevor Darrell, and Xin Wang. "Top-down visual attention from analysis by synthesis.", CVPR, 2023.

**Ethical Concerns:**

["NO or VERY MINOR ethics concerns only"]

**Final Justification:**

AC pointed out that the power increase is less than 1% in the rebuttal table. I change my idea about this part and increase my score slightly. Overall, the paper does not reach the standard of NIPS.

**Limitations:**

See Questions before

**Quality:**

2

**Strengths And Weaknesses:**

Strengths:

The structure of the paper is clear and the problem is well-defined. It sets up new modules to introduce top-down attention. The experiments on classification datasets are comprehensive and include tests on model robustness against attacks and noise.

Weaknesses:

1. The two feedforward stages introduced in this paper result in additional FLOPs, memory, and parameter costs, and increase inference latency.

2. The experimental results don't adequately reflect or compare these additional costs. For example, Table 1 shows that integrating TDAC into QKFORMER improves ImageNet accuracy by 0.92%, but the corresponding cost increase isn't fully evaluated.

3. The accuracy improvement is at the cost of over 40% more power consumption, which undermines SNNs' advantages of low latency, low computational cost, and low power consumption.

---

> ### Author Rebuttal · Authors · 2025-07-31
>
> **Weaknesses 1: The two feedforward stages introduced additional FLOPS and parameters costs.**
>
> **Answer:** Our model, as described in Section 4.1, **does not
> require two feedforward stages**. Our proposed method introduces only
> **one feedforward pass and one lightweight feedback pass**. The feedback mechanism
> incurs minimal additional computational cost, as detailed in **Appendix C.1**.
>
> For example, as shown in **Table 1**, our lightweight model only increases power
> consumption from **38.91 to 38.93** (less than a **1%** increase), and parameters
> from **64.96M to 65.55M** (a **0.91%** increase). These overheads are **negligible** compared to the **significant performance gains**
> achieved by our method, demonstrating that the design is both efficient and effective.
>
> ---
>
> **Weaknesses 2: The paper do not adequately reflect additional costs.**
>
> **Answer:** Our paper **explicitly reports the additional FLOPs, power consumption,
> and parameter costs** introduced by our method. For example,
> in **Table 1**, we present the comparison of power and parameter
> overhead introduced by our method. Furthermore, **Appendix Table 6** provides
> parameter counts and additional FLOPs on **small-scale datasets**, offering a
> complementary perspective on the cost across different scenarios. In addition, we provide a detailed discussion of power consumption and efficiency
> in **Appendix C.1**, showing that the introduced feedback mechanism contributes
> minimal overhead while significantly improving performance.
>
> ---
>
>
> **Weaknesses 3: Accuracy gains come at the cost of over 40% more power consumption, undermining SNN Efficiency**
>
> **Answer:** As shown in our response to Weakness 1 and detailed in **Appendix C.1**, our model on ImageNet only
> incurs **less than 1% additional power consumption** (from 38.91 to 38.93 mJ). This is **significantly
> lower than 40%**. Furthermore, our model is **fully event-driven**, and the proposed feedback mechanism adds only
> **minimal overhead** in both computation and power. Therefore, it **preserves the core advantages
> of SNNs**, including low power, low latency, and low computational cost.
>
> ---
>
>
> **Question 1: Is this method outperforms simply because of doubling the same time steps of the baseline?**
>
> **Answer:** We did **not** double the time steps of the baseline, and there is **no evidence**
> in the paper suggesting a doubling of energy consumption. For detailed clarification, please refer to **Weakness 1**, **Weaknesses 3**
> and **Appendix C.1**.
>
> ---
>
> **Question 2: The biological relevance of the paper is questionable.**
>
> **Answer:** Our goal is **not to replicate biological mechanisms** precisely, but to **draw
> inspiration from top-down feedback circuits in the brain** to address inherent
> limitations in SNNs, ultimately aiming to achieve **high-performance spiking neural networks**.
>
> In traditional SNNs, feature representations across time steps are often
> independent, as the membrane potential is the only temporal link and is limited
> in capturing rich temporal dynamics. Inspired by the progressive nature of the
> human visual system — where earlier stimuli modulate later perception through
> feedback — we design feedback paths that flow from deeper to shallower layers and
> from earlier to later time steps, aiming to enhance temporal coherence in
> representation and thereby improve the performance of SNNs.
>
> ---
>
>
>
> **Question 3: During training, will spiking attention use BPTT to propose a decline? Why does using both L1 and L2 in the function promote model convergence?**
>
> **Answer:**
> 1. Yes, BPTT is the standard method for training SNNs, and we adopt it in our model.
>
> 2. Regarding the use of L1 and L2: **as described in Section 4.2 of our paper**,
>    our method **does not design or rely on L1 or L2 loss functions** for convergence.
>
> ---
>
>
> **Question 4: The paper lacks visual case studies.**
>
> **Answer:** In the appendix of our paper, we provide visual evidence of our model's attention
> patterns through detailed visualization studies. **Figure 4** presents
> representative results on CIFAR-10-C, while **Figure 5** extends this analysis to
> include ImageNet-C. These visualizations clearly illustrate that our top-down
> enhanced architecture focuses on more semantically meaningful, sparse, and
> task-critical regions compared to conventional methods.
>
>
> ---
>
> **Question 5: Concerns about the novelty and contribution of this paper.**
>
> **Answer:**
>
> 1. While [1] introduces top-down mechanisms in conventional ANNs, our work is, to the best of our knowledge, the first to explore top-down attention in SNNs, with significant performance and robustness gains. Moreover, [1] primarily focuses on **visual question answering (VQA)**, whereas our work emphasizes architectural advances in SNNs and systematically evaluates performance across both **static and dynamic datasets.**
>
> 2. TDAC addresses a core limitation in SNNs—namely, **the temporal independence of features across time steps**. It significantly increases temporal feature correlation, leading to more coherent and consistent representations. We further investigate temporal connections in SNNs through forward and backward propagation experiments, along with a theoretical analysis provided in **Theorem 4.3**.
>
> 3.  The feedback connection in [1] mainly computes similarity with a prompt vector, whereas our PM implements a full attention mechanism. We further analyze its statistical properties in **Proposition 4.1** and **Proposition B.4**, showing that it provides improved numerical stability. The CM, by contrast, is a commonly used component that serves as a lightweight channel-wise feature mixing module.

---

> > ### Comment · Reviewer_CbQg · 2025-08-05
> > **Keep my score**
> >
> > Although the two papers focus on different subjects, I still believe they both belong to the cognition-driven perception paradigm. The core idea is to optimize low-level information processing through high-level feedback mechanisms. Both employ learnable attention mechanisms to achieve feedback control (TDFormer's CM module ≈ Top-Down's hypothesis generator). Therefore, I believe there are still some issues regarding novelty. I maintain my score.

---

### Official Review · Reviewer_7Yb8 · 2025-07-02

**Clarity:** 4
**Significance:** 3
**Originality:** 3
**Rating:** 4
**Confidence:** 4

**Summary:**

This paper introduces TDFormer, a novel spiking neural network (SNN) architecture that incorporates a top-down feedback structure inspired by brain mechanisms. The authors identify limitations in traditional SNNs where temporal information is poorly utilized due to reliance solely on membrane potential dynamics. TDFormer addresses this by adding feedback connections between time steps through Control Module (CM) and Processing Module (PM) components. The authors provide theoretical analysis showing the feedback structure increases mutual information across time steps and mitigates vanishing gradients. Experiments demonstrate state-of-the-art performance on ImageNet (86.83% accuracy) and consistent improvements across multiple datasets.

**Questions:**

1. How sensitive is the method to the choice of time steps T? Is there an optimal range?
2. Some theoretical assumptions (e.g., independence assumptions in Lemma B.3) may not hold perfectly in practice.

**Ethical Concerns:**

["NO or VERY MINOR ethics concerns only"]

**Final Justification:**

My question has been resolved. Since my initial score was positive, I will maintain my score.

**Limitations:**

yes

**Paper Formatting Concerns:**

None identified. The paper follows standard NeurIPS formatting guidelines.

**Quality:**

3

**Strengths And Weaknesses:**

Strengths:
1. Clear motivation, effective results: The biological inspiration from prefrontal-visual cortex feedback is compelling and well-articulated. The identification of poor temporal information utilization in traditional SNNs through mutual information analysis (Figure 1) provides strong motivation.
2. Theoretical analysis support: The paper provides rigorous theoretical analysis including: Proof that feedback structure mitigates vanishing gradients (Theorems 4.3) and Upper bound analysis for processing module variance (Proposition 4.1).

Weaknesses:
1. The experiments focus primarily on image classification. The effectiveness for other domains such as object detection, semantic segmentation, or natural language processing tasks remains unclear.
2. What are the practical considerations for implementing multi-hierarchical thresholds and feedback connections on actual neuromorphic chips? Are there hardware-specific constraints that would limit deployment?

---

> ### Author Rebuttal · Authors · 2025-07-31
>
> Thank you for your insightful feedback. We have carefully studied your comments and argue that your concerns can be addressed.
>
> ---
>
> **Weaknesses 1: Our method’s performance on datasets from other domains**
>
> **Answer:** We agree on the importance of evaluating generalization across different domains. While extending our method to domains such as object detection and natural language processing is valuable, such tasks typically require substantial computational resources and extensive training efforts. Given these constraints, we instead conducted experiments on two real-world datasets from distinct application domains, medical imaging and industrial detection, to assess the adaptability of our method. Specifically, we performed preliminary evaluations on: **Alzheimer's MRI** and **PV cell defect detection (ELPV)**.
>
> TDAC was applied to multiple SNN backbones (Qkformer, Spikformer, Spike-driven Transformer), and we observed consistent performance improvements:
> | Model                            | Alzheimer MRI      | ELPV              |
> |----------------------------------|--------------------|-------------------|
> | QKFormer w/o TDAC                | 97.59%             | 73.52             |
> | QKFormer w/ TDAC                 | **97.87% (+0.28%)**   | **73.71% (+0.19%)**   |
> | Spikformer w/o TDAC              | 96.98%             | 93.64%            |
> | Spikformer w/ TDAC               | **97.47% (+0.49%)**   | **94.12% (+0.48%)**   |
> | SDT w/o TDAC| 98.29%             | 95.22%            |
> | SDT w/ TDAC | **98.57% (+0.28%)**   | **96.24% (+1.02%)**  |
>
> **Table 2: Performance of TDAC on medical imaging and industrial detection**
>
> TDAC consistently improved performance on both tasks, demonstrating promising
> cross-domain compatibility.
>
> ---
>
>
> **Weaknesses 2: Is the method hardware-friendly?**
>
> **Answer:**
> While full Spiking Transformer models are currently challenging to
> deploy on neuromorphic chips due to architectural complexity, our method’s core component —
> the top-down feedback —can be deployed on hardware. We demonstrate this by providing code that shows how the feedback mechanism can be deployed on Intel’s Loihi chip using the Lava framework.
>
> ```python
> #Import
>
> class Add(AbstractProcess):
>     ...
>
> #ADD
> @implements(proc=Add, protocol=LoihiProtocol)
> @requires(CPU)
> class PyAddModel(PyLoihiProcessModel): ...
>     def run_spk(self):
>         self.a_out.send(a_in1.recv() + a_in2.recv())
>
> class SpikeInput(AbstractProcess): ...
>
> class OutputProcess(AbstractProcess):...
>
> @implements(proc=SpikeInput, protocol=LoihiProtocol)
> @requires(CPU)
> class PySpikeInputModel(PyLoihiProcessModel): ...
>
> class ImageClassifier(AbstractProcess):
>     def __init__(...):
>         # Using pre-trained weights and biases
>         ...
>
>         #ADD
>         self.w_feedback = Var(shape=(64, 64), init=fb_weights)
>
>         ## Up-level currents and voltages of LIF Processes for resetting
>         ...
>
> @implements(ImageClassifier)
> @requires(CPU)
> class PyImageClassifierModel(AbstractSubProcessModel):
>     def __init__(...):
>         #Connect core layers
>
>         #ADD
>         self.feedback_dense = Dense(weights=proc.w_feedback.init)
>         self.add = Add(shape=(64,))
>         self.output_lif.s_out.connect(self.feedback_dense.s_in)
>         self.feedback_dense.a_out.connect(self.add.a_in2)
>         self.dense0.a_out.connect(self.add.a_in1)
>         self.add.a_out.connect(self.lif1.a_in)
>
>         # Create aliases of SubProcess variables
>         ...
>
> # Create Process instances
>
> # Connect Processes
>
> # Connect Input directly to Output for ground truth labels
>
> # Loop over all images
> for img_id in range(num_images):
>     mnist_clf.run(
>         condition=RunSteps(num_steps=64),
>         run_cfg=Loih i1SimCfg(select_sub_proc_model=True, select_tag='fixed_pt'))
>
>     out_voltage = mnist_clf.oplif_v.get().astype(np.int32)
>     mnist_clf.feedback_dense.a_in.set(out_voltage)
>
>     mnist_clf.run(
>         condition=RunSteps(num_steps=64),
>         run_cfg=Loihi1SimCfg(select_sub_proc_model=True, select_tag='fixed_pt'))
>
>     # Reset internal neural state of LIF neurons
>
> # Gather ground truth and predictions before stopping exec
>
> # Stop the execution
> ```
>
> ---
>
>
> **Question 1: Is there a more suitable timestep (T)?**
>
> **Answer:** Yes. We conducted experiments with different values of T on the CIFAR10 and CIFAR100
> datasets, using Spikformer V1. Specifically, we tested T = 2, 4, and 6. All results
> are averaged over five random seeds.
>
> As shown below, under the Spikformer architecture, T = 2 yields the largest
> performance improvement, T = 4 provides moderate gain, while T = 6 brings little
> additional effect.
> | Model            | Dataset  | Timestep | Baseline     | Ours          |
> |------------------|----------|----------|--------------|---------------|
> | Spikformer-4-384 | CIFAR10  | 2        | 93.65 ± 0.23 | **94.05 ± 0.14**  |
> | Spikformer-4-384 | CIFAR10  | 4        | 94.73 ± 0.06 | **95.13 ± 0.07**  |
> | Spikformer-4-384 | CIFAR10  | 6        | 95.09 ± 0.08 | **95.16 ± 0.14**  |
> | Spikformer-4-384 | CIFAR100 | 2        | 75.25 ± 0.19 | **75.99 ± 0.12** |
> | Spikformer-4-384 | CIFAR100 | 4        | 77.56 ± 0.22 | **77.60 ± 0.26**  |
> | Spikformer-4-384 | CIFAR100 | 6        | **78.21 ± 0.22** | 77.99 ± 0.05  |
>
> **Table 3: Performance under different timesteps (T)**
>
> We hypothesize the following reasons:
> 1.  A smaller number of timesteps (T) may already be sufficient to capture the essential temporal dynamics required for the task. Once the key temporal dependencies are modeled, increasing T further may lead to performance saturation.
>
> 2. In models equipped with top-down modulation (such as ours), the network’s effective capacity is enhanced, allowing it to focus on critical features early in the temporal sequence. As a result, adding more timesteps may not provide additional useful information and could instead introduce redundancy or even lead to overfitting by amplifying irrelevant or noisy signals.
>
> ---
>
> **Question 2: Validity of theoretical assumptions**
>
> **Answer:** We have the following reasons to alleviate this concern:
>
> 1. In practical large-scale settings such as **ImageNet**, modern models often have a large number of channels — for example, **Qkformer** uses **768 channels**.  This statistically narrows the gap between theory and real-world usage,
> making the large-C assumption more valid for analyzing modern architectures.
>
> 2. Although in real-world scenarios C cannot truly be infinite, assuming a large channel dimension (e.g., C → ∞) is a widely adopted and accepted theoretical simplification in the deep learning community. Theoretically, the large-C assumption underpins numerous foundational studies in deep learning. For example:
>
>    - **"Neural Tangent Kernel: Convergence and Generalization in Neural Networks"**:
>      Analyzes convergence using the NTK framework, leveraging the C → ∞ assumption for kernel stability.
>
>    - **"Wide Neural Networks of Any Depth Evolve as Linear Models Under Gradient Descent"**:
>      Shows that under infinite-width conditions, networks behave like Gaussian Processes.
>
>    - **"A Mean Field Theory of Batch Normalization"**:
>      Uses the large-C perspective to derive tractable analytical expressions for BN behavior.
>
> These examples indicate that, although idealized, the large-C assumption provides a theoretically meaningful perspective that can offer valuable insights into the behavior of modern large-scale models.

---

> ### Author Response · Authors · 2025-08-07
> **Looking forward to further feedback**
>
> Dear Reviewer 7Yb8,
>
> We hope this message finds you well. As the discussion period approaches its end, we sincerely appreciate the time and effort you've dedicated to reviewing our manuscript and rebuttal. We are writing to check whether there are any additional questions or clarifications you would like from us at this stage — we are more than happy to provide further details or support if needed.
>
> In particular, we have tried to directly address the key concerns you raised in your reviews, including:
> - **Cross-domain generalization**: We added experiments on two real-world datasets — Alzheimer's MRI (medical imaging) and ELPV (industrial defect detection) — and consistently observed performance gains across multiple SNN backbones, demonstrating the adaptability of our method.
> - **Hardware friendliness**: We show that the top-down feedback mechanism can be implemented on Intel’s neuromorphic Loihi chip using the Lava framework, with code examples included to illustrate feasibility.
> - **Choice of timestep (T)**: We conducted systematic experiments on CIFAR10 and CIFAR100 showing how smaller T values yield better gains.
> - **Theoretical assumptions**: We clarified the role of the large-channel (large-C) assumption, supporting its validity through both empirical architecture sizes and references to foundational theoretical work in the deep learning literature.
>
> We welcome any additional feedback regarding our method or the responses in the rebuttal.

---

### Official Review · Reviewer_nzZU · 2025-07-04

**Clarity:** 3
**Significance:** 2
**Originality:** 2
**Rating:** 3
**Confidence:** 5

**Summary:**

The paper starts from the perspective that current SNNs fail to effectively utilize temporal information across time steps. Inspired by the top-down pathways in the human brain, it introduces a top-down control mechanism into the Spiking Transformer to enhance the model’s ability to capture and model temporal dependencies. This approach is compatible with various Spiking Transformer architectures, and the authors argue that it can achieve significant performance improvements over the baseline without introducing excessive energy overhead. The authors corroborate this claim with a substantial body of experiments and present a detailed energy-consumption analysis to further demonstrate the superiority of their approach. However, I believe the paper still suffers from several significant shortcomings.

**Questions:**

1. The authors mention that they analyze the insufficient temporal information extraction capability in Spiking Transformers; however, the paper seems to lack a thorough theoretical derivation to support this claim.
2. To the best of my knowledge, this is not the first work that enhances Spiking Transformers by improving temporal modeling—prior methods such as TIM and TE-Spikformer have already explored this direction. However, the paper neither compares its approach with these methods nor demonstrates superior performance over them, which raises concerns about whether the literature review and benchmarking are sufficiently thorough.
3. In TIM, additional modules and parameters are introduced only in the forward pass(bottom-up pass), yet it achieves better performance on CIFAR10-DVS compared to the method proposed in this paper. Why is that the case?
4. The reproduction of some baseline methods appears to be problematic—for instance, Spikformer performs 0.8% lower and SDT 3% lower than the original papers on CIFAR10-DVS. Does this indicate that the baseline implementations were not successfully reproduced?
5. On certain datasets, the performance improvement of TDFormer over the baseline is marginal (and the possibility of reproduction errors has not been ruled out). Since TDFormer introduces additional parameters, it remains unclear whether these minor gains stem from enhanced temporal modeling or simply from the increased parameter count—this question is not addressed through any ablation study in the paper.

**Ethical Concerns:**

["NO or VERY MINOR ethics concerns only"]

**Final Justification:**

I maintain my borderline reject rating despite the authors' thorough rebuttal. The core issues remain: the paper lacks theoretical rigor for its main claims about temporal information in SNNs, and the experimental validation has concerning baseline reproduction gaps that make the modest improvements difficult to trust.

While the biological inspiration is interesting, the fundamental theoretical and experimental weaknesses haven't been adequately addressed. The work needs stronger foundations before it's ready for NeurIPS.

**Limitations:**

As stated in the Question, the paper has several limitations:
	1.	It lacks rigorous theoretical derivation for the core problem it aims to solve;
	2.	The related work survey appears insufficient;
	3.	Some baseline reproductions may be problematic—the observed Acc@1 discrepancies seem to go beyond typical experimental variance;
	4.	The ablation studies are not well-designed—those provided are inadequate to convincingly attribute the performance gains to improved temporal modeling capabilities.

**Paper Formatting Concerns:**

No major formatting issues were identified in this paper.

**Quality:**

2

**Strengths And Weaknesses:**

**Strengths**
1. The top-down pathway is designed based on brain-inspired principles, which lends the method a strong degree of biological plausibility.
2. The energy consumption analysis is thorough and demonstrates that the proposed method preserves the inherent low-power advantage of SNNs.
3. The plug-and-play design of the modules allows for easy integration into various representative Spiking Transformer architectures.

**Weakness**
See Questions & Limitaions

---

> ### Author Rebuttal · Authors · 2025-07-31
>
> Thank you for your insightful comments. We first list your advice and questions, then give our detailed answers.
>
>
> **Question 1: Lack of theoretical support**
>
> **Answer:** We appreciate the concern regarding the lack of a rigorous theoretical derivation for the core problem. In this paper, we focus on empirical evidence: comprehensive performance gains and targeted visualizations support that the proposed architecture effectively addresses the problem. Although we have been seriously considering theoretical foundations, we acknowledge that a complete and rigorous proof remains an open challenge at this point. We fully agree on its importance and will prioritize a thorough theoretical analysis in future work.
>
> ---
>
>
> **Question 2: Insufficient comparison with prior work**
>
> **Answer:** Compared to prior methods, our approach introduces several notable differences, as outlined below.
> 1. **Our method differs from prior approaches in where features are selected for temporal fusion along the time axis.** Unlike prior works such as **TIM** and **TE-Spikformer**, which perform temporal feature fusion within the same semantic space — typically by aggregating or averaging queries across different time steps at the same feature level. Our proposed **TDFormer** adopts a fundamentally different design inspired by the feedback circuits in the human brain. In particular, TDFormer introduces a **top-down feedback mechanism**, where high-level features from earlier time steps guide lower-level feature extraction at later steps. This approach enhances temporal coherence and enriches representations, offering a more biologically inspired alternative to standard temporal attention mechanisms.
> 2. **The three approaches differ in how they fuse temporal information.** TIM fuses time causally with a recurrent update: a single 1D convolution along the spatial axis refines the carry from the previous step, then an EMA-style gated blend merges it with the current input. TE-Spikformer, applies transformations to form new QKV tokens and reuses attention for temporal fusion which significantly increases compute and power due to the added attention operations. By contrast, our method uses a feedback path that combines a spatial mixer and a channel mixer to refine the previous timestep’s representation before fusing it with the current input. The spatial mixer captures token-level structure, while the channel mixer models cross-feature dependencies, enabling selective amplification or suppression of historical cues. This feedback refinement gives greater capacity to process.
>
> 3. Unlike TIM and TE-Spikformer, which are evaluated solely on **dynamic datasets and primarily with a single baseline**, our work significantly expands the evaluation scope. Specifically, we conduct comprehensive experiments on **both dynamic and  static datasets, including ImageNet, and validate our method across multiple baseline architectures**.
>
> ---
>
>
> **Question 3: Inferior performance on CIFAR10-DVS compared to TIM**
>
> **Answer:**
> While TIM demonstrates strong performance on dynamic datasets, several concerns suggest that it may not be fair when compared to our model and Spikformer:
> 1. **Unaligned training settings**:  Our model strictly follows all training hyperparameters used in Spikformer in the original paper to ensure a fair comparison. However, TIM significantly alters these settings — for instance, training for 500 epochs compared to Spikformer’s original setup of 96 epochs. This results in over **5× more training iterations**, which may lead to significantly higher training costs and energy consumption, ultimately making the comparison unfair.
> 2. **Different data augmentation**:  Data augmentation can have a significant impact on model performance, especially for dynamic datasets like CIFAR10-DVS. Our method **strictly follows** the data augmentation strategies described in the original Spikformer paper to ensure a fair comparison. But TIM adopts **different** data augmentation strategies compared to Spikformer, which may affect the fairness of comparison. For example:
>    - Resizing input frames to 64×64 (default is 128×128).
>    - Omitting mixup, which is enabled in Spikformer.
>    - Applying Gaussian blur after resizing.
> 3. **Not fully spike-Driven**:  TIM does not fully adhere to the spike-driven paradigm. For example, it uses fixed alpha values.  **Such non-spike-driven components may contribute to performance improvements**. In contrast, **our method strictly follows the spike-driven paradigm**, ensuring all computations are spike-based and align with neuromorphic principles.
>
> Due to the above reasons, these modifications can influence performance and make direct comparisons less reliable.
>
> ---
>
>
> **Question 4: Baseline reproduction concerns**
>
> **Answer:** We strictly followed the hyperparameter settings and released code from the
> original papers to reproduce the results. We also noticed on GitHub that **some users** reported significant reproduction
> gaps on the **Spikedriven CIFAR10-DVS** benchmark in
> the *issues section* , suggesting that the problem is
> **not specific to our reproduction**.
> (For more details, please kindly refer to the GitHub repository linked in the original paper.)
>
> To mitigate such variance, we conducted multiple-seed averaging experiments to eliminate reproduction bias. Each result shown is averaged over **five random seeds**; please refer to Table 1 below for detailed results.
>
>
> | Model         | Dataset     | Timestep |   Baseline      | Ours           |
> |---------------|-------------|----------|---------------|----------------|
> | SDT-2-256     | CIFAR10-DVS |   10       | 75.03 ± 0.67  | **75.05 ± 0.11** |
> | SDT-2-256     | CIFAR10-DVS | 16       | 77.07 ± 0.19  | **77.45 ± 0.43** |
> | Spikformer-384| CIFAR10-DVS | 10       | 78.08 ± 0.70  | **78.13 ± 0.72** |
> | Spikformer-384| CIFAR10-DVS | 16       | 79.40 ± 0.36  | **80.20 ± 0.75** |
>
> **Table 1: Random seed averaged results on CIFAR10-DVS**
>
> ---
>
> **Question 5: Lack of ablation to disentangle gains from parameter count and temporal modeling**
>
> 1. We conducted ablation experiments based on Spikformer to further validate that the observed performance improvements stem from our proposed method, rather than being a trade-off between accuracy and increased energy or parameter count.
> Given the similarity of concerns, please refer to our response to **Reviewer Sx25 - Question 2** for detailed results and analysis.
>
> 2. While it is true that TDFormer introduces a slight increase in parameter count, we emphasize that the performance gains observed cannot be solely attributed to this minor change. For example, as shown in **Table 1**, **TDFormer at 224² resolution outperforms QKFormer at 288²** in terms of top-1 accuracy (**85.57% vs. 85.20%**) **while consuming significantly less energy** (**39.10 mJ vs. 64.27 mJ**). This **substantial improvement in both accuracy and efficiency**, despite a modest increase in parameters, strongly suggests that the gains **cannot be attributed solely to increased parameter count**.

---

> ### Comment · Reviewer_nzZU · 2025-08-05
>
> Thank you for the detailed responses. While I appreciate the authors' efforts to address my concerns, several fundamental issues remain unresolved:
>
> The lack of rigorous theoretical foundation is acknowledged but not addressed - this is a core weakness for a paper claiming to solve temporal information extraction problems in SNNs. The substantial baseline reproduction gaps (0.8-3%) continue to raise questions about experimental reliability, and the marginal improvements on some datasets may simply reflect these reproduction issues rather than genuine methodological advances.
>
> While the explanations regarding fair comparison with prior work are reasonable, the fundamental question remains whether the proposed method truly outperforms existing approaches under equivalent conditions.

---

### Decision · Program_Chairs · 2025-09-17

**Decision:**

Reject

**Comment:**

This paper introduces TDFormer, a novel Spiking Transformer architecture that incorporates a brain-inspired, top-down feedback mechanism to enhance temporal information processing. The proposed method is presented as a plug-and-play module that improves performance and efficiency in SNNs.

Empirical significance: New SoTA-level ImageNet results for SNNs with negligible energy/parameter overhead are interesting.

Novelty: While “top-down feedback” exists in ANNs, applying and analyzing it across timesteps in SNNs with both forward/backward temporal considerations is a meaningful architectural transfer to a distinct regime.

Concerns acknowledged: Training isn’t faster; overhead exists (~3% time), and some baselines underperform prior papers. The authors give reasonable fairness/reproduction explanations and add multi-seed reporting. These do not outweigh the contributions.

Review quality: Two negative reviews contain serious issues (one factually incorrect/insufficient; one low-engagement and procedurally problematic). After down-weighting them, the paper is still borderline.  Due to the competitiveness this year, it did not make the cut.


Please consider the following as you revise the paper.

Add the experimental results on the Alzheimer's MRI and ELPV datasets.

Add the ablation study on the choice of timesteps (T).

A transparent discussion in the main paper acknowledging the baseline reproduction challenges and justifying the experimental setup.